# SpecTr: Fast Speculative Decoding via Optimal Transport

**Ziteng Sun**[*]
Google Research, New York
zitengsun@google.com

**Ananda Theertha Suresh**[*]
Google Research, New York
theertha@google.com

**Jae Hun Ro**
Google Research, New York
jaero@google.com

**Ahmad Beirami**
Google Research, New York
beirami@google.com

**Himanshu Jain**
Google Research, New York
himj@google.com

**Felix Yu**
Google Research, New York
felixyu@google.com

## Abstract

Autoregressive sampling from large language models has led to state-of-the-art results in several natural language tasks. However, autoregressive sampling generates tokens one at a time making it slow, and even prohibitive in certain tasks. One way to speed up sampling is *speculative decoding*: use a small model to sample a *draft* (block or sequence of tokens), and then score all tokens in the draft by the large language model in parallel. A subset of the tokens in the draft are accepted (and the rest rejected) based on a statistical method to guarantee that the final output follows the distribution of the large model. In this work, we provide a principled understanding of speculative decoding through the lens of optimal transport (OT) with *membership cost*. This framework can be viewed as an extension of the well-known *maximal-coupling* problem. This new formulation enables us to generalize the speculative decoding method to allow for a set of $k$ candidates at the token-level, which leads to an improved optimal membership cost. We show that the optimal draft selection algorithm (transport plan) can be computed via linear programming, whose best-known runtime is exponential in $k$. We then propose a valid draft selection algorithm whose acceptance probability is $(1 - 1/e)$-optimal multiplicatively. Moreover, it can be computed in time almost linear with size of domain of a single token. Using this new draft selection algorithm, we develop a new autoregressive sampling algorithm called *SpecTr*, which provides speedup in decoding while ensuring that there is no quality degradation in the decoded output. We experimentally demonstrate that for state-of-the-art large language models, the proposed approach achieves a wall clock speedup of 2.13X, a further 1.37X speedup over speculative decoding on standard benchmarks.

## 1  Introduction

Autoregressive language models have shown to achieve state-of-the-art results in several natural language tasks [2, 5, 25, 26]. During inference, given a context $x^t := x(1), x(2) \ldots, x(t)$, an autoregressive model $\mathcal{M}_b$ generates successive tokens $x(t+1), x(t+2), \ldots$ via temperature sampling [1, 10], where the next token $x(t + 1)$ is drawn from the temperature-scaled distribution $\mathcal{M}_b(\cdot|x^t)$. If the temperature is zero, i.e., greedy decoding, the next token is determined by the maximum likelihood method i.e., $x(t+1) = \arg\max_{x \in \Omega} \mathcal{M}_b(x|x^t)$, where $\Omega$ is the domain of a single token also referred to as the vocabulary. The sampling approach can be further combined with other sampling primitives such as nucleus sampling [16] and top-$k$ sampling [9, 22].

---

[*]Equal contribution.

37th Conference on Neural Information Processing Systems (NeurIPS 2023).

All these approaches are autoregressive decoding[2] methods, where tokens are generated serially one after another, which can be slow or even prohibitive in several applications [23]. Hence, several techniques have been proposed to improve the speed of decoding. Before we proceed further, we first present some notations and a simplified computational model.

**Notations.** We use $x^{i:j}$ to denote the sequence $x(i), x(i+1), \ldots, x(j)$ and when $i = 1$, we simply use $x^j = x^{1:j}$. $x(i)$ denotes the $i$-th entry of $x$. Subscripts are used to distinguish between different sequences, *e.g.,* $x_1^t$ and $x_2^t$ denote two sequences of length $t$. We use $[n]$ to denote the set $\{1, \ldots, n\}$.

**A simplified computational model.**

- **Standard inference.** Given a context $x^t$, with $O(t^2)$ computation and $O(1)$ time, an autoregressive model $\mathcal{M}_b$ can compute $\mathcal{M}_b(y|x^t)$, the (temperature-scaled) probability of all possible next tokens $y \in \Omega$.
- **Parallelization along the time axis.** Given a context $x^t$, with $O(t^2)$ computation and $O(1)$ time, an autoregressive model $\mathcal{M}_b$ can compute $\mathcal{M}_b(y|x^i)$, for all $y \in \Omega$ and $i \in \{1, 2, \ldots, t\}$.
- **Parallelization along time and batch axis.** Let $K$ be the maximum batch size that can be used during the inference of the autoregressive model. Given several contexts, $x_1^t, x_2^t, \ldots x_K^t$, with $O(Kt^2)$ computation and $O(1)$ time, an autoregressive model $\mathcal{M}_b$ can compute $\mathcal{M}_b(y|x_j^i)$, for all $y \in \Omega$, $i \in [t]$, and $j \in [K]$.[3]

The above computation model shows that parallelizing along time and batch axes does not increase the computation time. It is a simplified characterization of the typical hardware, such as TPUs and GPUs, used in neural network inference. Previous approaches also assume similar computational model to devise faster decoding algorithms [19, 4]. In practice, there will be some overhead depending on hardware, implementation and resource utilization. In Appendix E, we experimentally verify that the theoretical gains are largely preserved for a large transformer model in practice. We also note that there are efficient transformer architectures, which reduces the computation cost from $O(t^2)$ to $O(t \log t)$ (see [24] for a detailed survey). Such approaches are orthogonal to the focus of this paper, and they can be easily combined with our approach.

Broadly speaking, multiple previous approaches proposed to guess a few possible future tokens using an efficient model. They then compute several conditional probability distributions from the large model based on the guesses. Computing the distributions takes $O(1)$ time due to parallelization along the time axis. The guessed tokens are then accepted or rejected based on a statistical method such that the accepted tokens are effectively samples from the large model. This guarantees that there is provably no degradation in the quality of the decoded output compared to that of the large model. When the guesses are plausible under the large model, multiple tokens will be accepted, leading to a larger gain in latency improvement. We will further characterize the acceptance probability as a function of the closeness of the distributions of large model and the small model. While this approach incurs the same computation cost as vanilla decoding (under the simplified computational model assumed in this paper), it can significantly improve decoding latency due to parallelization.

The goal of this work is to provide a principled understanding of the above approaches and discuss optimality conditions and algorithmic improvements. We start by providing a more formal overview of speculative decoding and related works.

## 2 Previous works and speculative decoding

Previous approaches make use of parallelization along the time axis to provide speedups. They first predict multiple tokens and validate if these multiple tokens can be generated by the model with the corresponding sampling or decoding scheme. For greedy decoding, multiple tokens can be predicted by a separate model [23], aggressive decoding [11], or retrieval augmented text [28]. For sampling, recently [19, 4] proposed an algorithm called speculative decoding, and we provide an overview of this algorithm in the rest of the section. Suppose we have access to a computationally-inexpensive draft model $\mathcal{M}_s$, which predicts the next token given the context, and the predictions of $\mathcal{M}_s$ are

---

[2]In this work, we use the words *sampling* and *decoding* interchangably to refer to the process of sequentially generating tokens from a language model.

[3]When the assumption holds, one could naively batch multiple decoding contexts, which improves decoding throughput, but not the latency of each context.

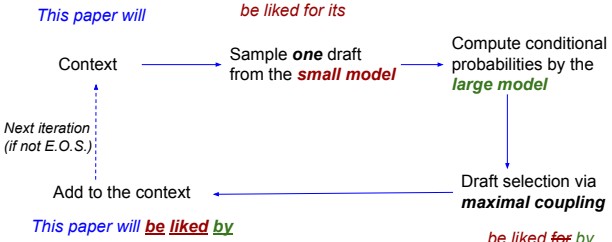

Figure 1: One iteration of speculative decoding [19, 4]. Tokens in blue are decoded tokens from previous iterations, which are used as context for the current iteration. Tokens in red are drafts from the small model based on the context. The underlined tokens are the newly decoded tokens in the current iteration, where underlined red tokens represent tokens selected from the draft and underlined green token is selected from the residual distribution.

close to that of $\mathcal{M}_b$ for most contexts. Suppose we have obtained prefix $x^t$. The next iteration of the speculative algorithm can be broken down into three steps (see Fig. 1 for an illustration).

1. **Draft construction.** The draft model is used to efficiently and "speculatively" sample $L$ tokens, $\tilde{x}(t+1), \ldots, \tilde{x}(t+L)$. We keep the conditional probabilities on the next token $\mathcal{M}_s(y \mid x^t, \tilde{x}^{t+1:t+i})$ for each $i < L$ and $\forall y \in \Omega$.
2. **Conditional probability computation.** After observing the samples, we compute the conditional distributions $\mathcal{M}_b(y \mid x^t, \tilde{x}^{t+1:t+i})$ for each $i \leq L$ and $\forall y \in \Omega$ in parallel (along time axis) in $O(1)$ time.
3. **Draft selection.** Validate and select first $L'$ of the $L$ tokens and set $x(t+i) = \tilde{x}(t+i)$ for $i \leq L'$ given the draft sequence and the conditional probabilities from both models. Sample a token $\tilde{x}(t+L'+1)$ from a *residual* distribution as a *correction* to the rejected token.[4]

After this step, we use $x_1^{t+L'+1}$ as the next context and sample the next few tokens using speculative decoding iteratively. For a complete statement of the algorithm, we refer the readers to [19, 4]. The crux of the above steps is draft selection, which given a draft sequence and the conditional probabilities from both models, selects a valid sequence such that the output has the same distribution as that of the large model. In speculative decoding, this is achieved via recursively applying a token-level maximal coupling algorithm, which is provided in Algorithm 1. Note that for the draft selection, Algorithm 1 is applied where $p$ is the conditional distribution of the draft model $\mathcal{M}_s(\cdot \mid x^t)$ and $q$ is the conditional distribution of the large model $\mathcal{M}_b(\cdot \mid x^t)$ (which may be further conditioned on the newly decoded tokens).

---

**Algorithm 1** Token-level maximal coupling

---

**Input:** Distributions $p, q$, Draft sample $X \sim p$.
1: Compute the *residual* distribution $p^{\text{res}}$ where $\forall x \in \Omega, p^{\text{res}}(x) = \frac{q(x) - \min\{p(x), q(x)\}}{1 - \sum_{x'} \min\{p(x'), q(x')\}}$.
2: Sample $\eta \sim U(0, 1)$.
3: **if** $\eta \leq \min\left(1, \frac{q(X)}{p(X)}\right)$ **then**
4:     **Return** $Y = X$. {*Accept the draft token.*}
5: **end if**
6: **Return** $Y \sim p^{\text{res}}$. {*Sample a corrected token from the residual distribution.*}

---

Algorithm 1 returns a random variable $Y$ which either is the accepted input $X$ or a sample from the residual distribution $p^{\text{res}}$, which is defined in Step 1 of Algorithm 1. The algorithm is recursively applied as long as the draft tokens are accepted to select the first $L' \leq L$ tokens from the draft model. For the first rejected token, the sample $Y$ from the residual distribution is used as a *correction*. Previous works showed that if $X \sim p$, then $Y \sim q$ [19, 4]. In the case of the draft selection, this means that the output of the algorithm is distributed according to $\mathcal{M}_b(\cdot \mid x^t)$, which is exactly the

---

[4]See Algorithm 1 for definition of the residual distribution. When $L' = L$, no token is rejected. The residual will just be the conditional probability $\mathcal{M}_b(\cdot \mid x^{t+L})$, which gives an *extra* decoded token.

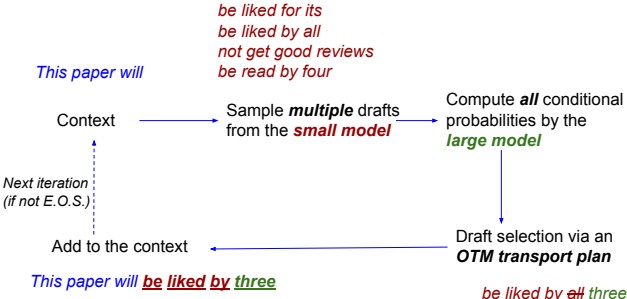

Figure 2: One iteration of *SpecTr*. Tokens in blue are decoded tokens from previous iterations, which are used as context for the current iteration. Tokens in red are drafts from the small model based on the context. The underlined tokens are the newly decoded tokens in the current iteration, where underlined red tokens represent tokens selected from the draft and underlined green token is selected from the residual distribution. See Fig. 3 for a more detailed run of the draft selection step.

desired outcome. Furthermore

$$\Pr(Y = X) = \sum_{x \in \Omega} \min(p(x), q(x)) = 1 - d_{\mathrm{TV}}(p, q),$$

where $d_{\mathrm{TV}}$ is the total variation distance between $p$ and $q$. The closer $p$ and $q$ are in $d_{\mathrm{TV}}$, the higher the chance of $\Pr(Y = X)$, and fewer the number of serial calls to the larger model. In the ideal case, if $p = q$, then $\Pr(Y = X) = 1$, i.e., the draft token is always accepted, and when used for speculative decoding we have $L' = L$. Together with the extra sampled token[5] from $\mathcal{M}_b$, $L + 1$ tokens are obtained in one iteration. In such a case, based on our computational model (Section 1), assuming the decoding time of draft model is negligible, the speedup is $(L + 1)$ times.

## 3 Our contributions

From a theoretical viewpoint, the speculative decoding algorithm raises multiple questions.

- What is the relationship between speculative decoding and the broader literature of sampling in statistics?
- Is speculative decoding optimal in an information-theoretic sense?
- Speculative decoding uses parallelization along time to speed up decoding; would it be possible to use parallelization along batch (number of drafts) to further improve decoding speed?

We provide answers to all the above questions in this work. We first relate the problem of speculative decoding to the broader and well-studied discrete optimal transport theory through a token-level coupling problem (Section 4). With this connection, it becomes clear that the token-level draft selection is the optimal solution for optimal transport with indicator cost function and also related to the problem of maximal coupling [8]. Based on the connection to optimal transport, we show that one can further speed up the decoding by parallelizing along the batch axis by using multiple drafts from the draft model (Section 5).

More precisely, we formulate the token-level draft selection problem as a discrete optimal transport problem with membership cost, which is referred to as OTM. Discrete optimal transport can be solved with a linear program, but the number of variables is exponential in batch size, which can be prohibitive. To address this, we propose a valid transport plan that can be efficiently computed. Moreover, it achieves a $(1 - 1/e)$-approximation of the optimal acceptance probability (Section 6).

With the theoretically motivated algorithms and guarantees, we circle back to speeding up decoding and propose a new algorithm called *SpecTr* and theoretically show that it can be used to derive valid sequences from the large model with better speedups (Section 7). See Fig. 2 for an illustration of *SpecTr*. Compared to speculative decoding (Fig. 1), the main difference lies in the number of sampled drafts sampled from the small model and the selection algorithm that selects a valid sequence from multiple draft sequences. We remark here that the latter requires completely new statistical

---

[5]When $L' = L$, $x(t + L + 1)$ is sampled from $\mathcal{M}_b(\cdot \mid x^{t+L})$.

| | be | liked | for | its |
| This paper will | be | liked | by | ~~all~~ three |
| | not | get | good | reviews |
| | be | read | by | four |

Figure 3: An example run of the sequence-level draft selection in *SpecTr* with $L = 4$ and 4 draft sequences. In the first step, there are 4 drafts tokens, and the token-level draft selection algorithm selects the word '*be*' which appeared thrice. Note that all tokens following '*be*' are valid draft tokens from the small model. In the second step, there are 3 drafts and the selection algorithm selects '*liked*'. The next token-level selection algorithm will have two drafts ('*by*' and '*for*') and it selects '*by*'. Finally, there is only one draft following '*by*', and the selection algorithm doesn't select it and outputs '*three*' as a correction. The process ends and a total of 4 tokens are generated.

tools, and the connection between the token-level draft selection and OTM is critical for obtaining valid transport plans with good guarantees. We view this as one of the main contributions of the work. Similar to speculative decoding, there is provably no degradation in the quality of the decoded output compared that of the large model.

We then experimentally demonstrate the benefit of our approach on standard datasets (Section 8). More precisely, we show that for state-of-the-art large language models, *SpecTr* achieves a wall clock speedup of 2.13X, a further 1.37X speedup over speculative decoding on standard benchmarks.

## 4   Token-level draft selection and optimal transport

In this section, we focus on the draft selection step of *SpecTr*. We start by considering the case when $L = 1$, which is a token-level draft selection problem. In particular, given context $x^t$, let $X_1, \ldots X_k$ be a collection of draft tokens sampled from the small model, *e.g.,* sampled *i.i.d.* from $\mathcal{M}_s(\cdot \mid x^t)$. Note that by our assumption of the computation model, we could compute the following conditional probabilities from the large model in parallel ( along time and batch axes):

$$\mathcal{M}_b(\cdot \mid x^t) \qquad \text{and} \qquad \forall i \in [k], \quad \mathcal{M}_b(\cdot \mid x^t, X_i).$$

The goal of the draft selection algorithm $f : \Omega^k \to \Omega$ is to output $Y = f(X^k)$, whose distribution follows $\mathcal{M}_b(\cdot \mid x^t)$, and hence is a valid sample from the large model. Moreover, when $Y \in \{X_1, \ldots, X_k\}$, we could sample an extra token from $\mathcal{M}_b(\cdot \mid x^t, Y)$ without calling $\mathcal{M}_b$ since we have already computed the conditional probabilities $\mathcal{M}_b(\cdot \mid x^t, Y)$. Hence we would like to maximize the probability that we accept one token from the set of drafts.

When $L > 1$, the drafts are sequences sampled from $\mathcal{M}_s$, a sequence of token-level draft selection algorithms could be used along the time axis to select a valid sequence from the $\mathcal{M}_b$. See an example in Fig. 3. The full details about the sequence-level selection algorithm is provided in Section 7.

The reminder of the section will be focused on the token-level draft selection problem. From the above discussion, there are the two main goals of the draft selection problem.

- **Validity.** The output token is always a valid token from the large model *i.e.,* its distribution follows the conditional probability of the large model. This guarantees that there is no quality degradation compared to the large model.

- **Maximizing acceptance.** The higher the probability that we accept a draft token, the more serial computation we can save through parallelization, and hence better speedup.

Before proposing our framework to achieve the above goals, we would like to first discuss the technical challenge of draft selection with multiple draft tokens. One attempt is to sequentially apply the acceptance phase of Algorithm 1 (line 3 - 5) to each draft token $X_i$ with $p = \mathcal{M}_s(\cdot \mid x^t)$ and $q = \mathcal{M}_b(\cdot \mid x^t)$. However, this approach would not guarantee that the final accepted token is from the desired distribution. To see this, consider the example of $p = \text{Ber}(1)$ and $q = \text{Ber}(1/2)$.[6] Then we have $\forall i = 1, \ldots, k$, $X_i = 1$ and each of them will be accepted with probability $1/2$. After

---

[6] Ber$(b)$ denotes a Bernoulli distribution with the probability of seeing a head $b$.

applying Algorithm 1 to all $X_i$'s, the probability of getting a 1 will be at least $1 - 1/2^k$ and hence the output distribution would not be $\text{Ber}(1/2)$ for $k > 1$. Therefore the algorithm does not produce valid samples, which is a requirement of the draft selection problem.

In this work, we conduct a principled investigation of the draft selection problem, and show that these two main goals could be captured by the framework of optimal transport with a properly defined cost function. Next we define optimal transport formally and then connect it to draft selection with one draft. The generalization to multiple drafts is provided in Section 5.

**Coupling and optimal transport.** To simplify notations, we assume $\Omega$ is a discrete domain.

**Definition 1** (Coupling). *For two probability distributions $P$ over $\mathcal{X}$ and $Q$ over $\mathcal{Y}$, we say a joint distribution $\pi$ supported over $\mathcal{X} \times \mathcal{Y}$ is a coupling between $P$ and $Q$ if $\forall x, y, \pi(x, y) \geq 0$,*

$$\forall y \in \mathcal{Y}, \ \sum_{x \in \mathcal{X}} \pi(x, y) = Q(y), \qquad \text{and} \qquad \forall x \in \mathcal{X}, \ \sum_{y \in \mathcal{Y}} \pi(x, y) = P(x).$$

We use $\Pi(P, Q)$ to denote the set of all possible couplings between $P$ and $Q$.

When it is clear from context, we will overload notation and refer to the probabilistic mapping $f_\pi : \mathcal{X} \to \mathcal{Y}$ introduced by the conditional probability $\pi(y \mid x) := \pi(x, y)/P(x)$ as a coupling, which is also referred to as a transport plan from $P$ to $Q$ [27]. In this paper, we will set $P$ to be the distribution of the draft tokens and $Q$ to be the target distribution of the output token. In this case, the $f_\pi$ is a valid draft selection algorithm. Formally, this is stated in the claim below.

**Claim 1.** *For all $\pi \in \Pi(P, Q)$, let $f_\pi$ be the probabilistic mapping defined above . If $X \sim P$, then $f_\pi(X) \sim Q$.*

In this paper, we will design selection algorithms by finding valid couplings between the draft distribution and target distribution to guarantee validity of the output tokens.

**Definition 2** (Optimal Transport (OT) [27]). *For a cost function $c : \mathcal{X} \times \mathcal{Y} \to \mathbb{R}_+$, the transportation cost of a coupling is defined as:*

$$C(\pi) = \mathbb{E}_{X,Y \sim \pi} \left[ c(X, Y) \right].$$

The *optimal transport plan* is the coupling $\pi \in \Pi(P, Q)$ that minimizes the transportation cost.

**Speculative decoding with one draft token.** With these definitions in place, we can see that with $\mathcal{X} = \mathcal{Y} = \Omega$, the domain of the tokens and $P = p, Q = q$, we recover the speculative decoding objective with one draft token using the cost function of *indicator cost*, which captures the resampling cost, defined below:

$$\forall x \in \Omega, \ y \in \Omega, \qquad c(x, y) = \mathbb{1} \left\{ y \neq x \right\}.$$

The transportation cost of the coupling will be $C(\pi) = \mathbb{E}_{X,Y \sim \pi} \left[ \mathbb{1} \left\{ Y \neq X \right\} \right] = \mathbb{P}_{X,Y \sim \pi}(Y \neq X)$. This optimal transport cost is known to be

$$\min_{\pi \in \Pi(p,q)} \mathbb{P}_{X,Y \sim \pi}(Y \neq X) = \sum_{x \in \Omega} \min(p(x), q(x)), \tag{1}$$

which is achieved by the maximal coupling between $p$ and $q$ stated in Algorithm 1 [8]. And hence speculative sampling achieves the optimal cost with one draft token.

# 5 Optimal transport with multiple draft tokens

In this section, we generalize token-level selection to allow for multiple drafts. More formally, let $\mathcal{X} = \Omega^k$ for some $k \in \mathbb{N}_+$, which is the space of $k$ draft tokens from $\Omega$ and $\mathcal{Y} = \Omega$, which is the space of the final sampled token from the desired distribution. To characterize the resampling cost with multiple draft tokens, we use the cost function of *membership cost*, defined below:

$$\forall x \in \Omega^k, \ y \in \Omega, \qquad c(x, y) = \mathbb{1} \left\{ y \notin S(x) \right\},$$

where $S(x) = \{ o \in \Omega \mid o \text{ appears in } x \}$ denotes the set of distinct elements in $x$. When $k = 1$, it recovers the indicator cost mentioned before. The transportation cost of the coupling is

$$C(\pi) = \mathbb{E}_{X,Y \sim \pi} \left[ \mathbb{1} \left\{ Y \notin S(X) \right\} \right] = \mathbb{P}_{X,Y \sim \pi}(Y \notin S(X)). \tag{2}$$

We will also refer to the above cost $C(\pi)$ as the *rejection probability* due to its probabilistic interpretation. And similarly, $\alpha(\pi):=1 - C(\pi) = \mathbb{P}_{X,Y\sim\pi}(Y \in S(X))$ will be the *acceptance probability*.

From now on we will use membership cost as the default cost function and refer to the optimal transport solution as *optimal transport with membership cost* (OTM). We use $\pi^*$ to denote the coupling that minimizes this cost $\pi^* = \arg\min_{\pi\in\Pi(P,Q)} C(\pi)$;[7] and the cost $C(\pi^*)$ is referred to as the *optimal transport cost* between $P$ and $Q$. We use $\alpha(P,Q) = 1 - C(\pi^*)$ to denote the corresponding optimal acceptance probability.

**Draft selection with *i.i.d.* draft tokens.** In this paper, we will mainly focus on the case when the draft tokens are *i.i.d.* samples from a base distribution.[8] Let $p, q$ be supported over $\Omega$ and the goal is to obtain one valid token from $q$ given $k$ *i.i.d.* samples from $p$. For *SpecTr* with context $x^t$, we have $p = \mathcal{M}_s(\cdot \mid x^t)$ and $q = \mathcal{M}_b(\cdot \mid x^t)$. We set $P = p^{\otimes k}$, a product distribution whose marginals are all $p$, and $Q = q$. The OT problem we want to solve is the following:

$$\min C(\pi) \quad s.t. \quad \pi \in \Pi(p^{\otimes k}, q). \tag{3}$$

We overload notation and denote the *optimal acceptance probability* as $\alpha_k(p,q):=\alpha(p^{\otimes k}, q) = 1 - C(\pi^*)$. To better understand the quantity, we state a few properties about $\alpha_k$.

**Lemma 1.** *(Appendix A.2) The optimal acceptance probability statisfies the following properties.*

- ***Monotonicity.** For any $p, q$ and $k \geq 1$, $\alpha_k(p,q) \leq \alpha_{k+1}(p,q)$.*

- ***Consistency.** If $\forall x \in \Omega$, $q(x)/p(x)$ is bounded, we have $\lim_{k\to\infty} \alpha_k(p,q) = 1$. Else, $\lim_{k\to\infty} \alpha_k(p,q) = \sum_{x\in\Omega} \mathbb{1}\{p(x) > 0\} q(x)$.*

The above properties demonstrate that for a large $k$, the value of $\alpha_k$ can become large. Hence increasing $k$ could increase the acceptance probability, leading to further speedups. We now focus on computing the optimal transport plan and the optimal acceptance probability.

**OTM via Linear programming.** Optimal transport in discrete domain has been studied extensively [17, 21, 14], and it is shown that the optimal transport problem is equivalent to the following linear programming problem:

$$\min \sum_{x\in\Omega^k} \sum_{y\in\Omega} \pi(x,y)\mathbb{1}\{y \notin S(x)\} \qquad s.t. \quad \pi \in \Pi(P,Q). \tag{4}$$

The linear program in (4) has $|\Omega|^{k+1}$ variables and $|\Omega|^k + |\Omega|$ equality constraints (see Definition 1). Linear programming can be solved in time polynomial in the number of variables and constraints [7, 21, 18],[9] implying the following lemma.

**Lemma 2.** *Given $p, q$ over $\Omega$, the solution to Eq. (3) can be computed in time $O(|\Omega|^{O(k)})$.*

We refer to the optimal coupling obtained above as OTM-$k$ and denote it as $\pi^{\mathrm{OTM}-k}$. When $k = 1$, there is a closed form expression for the optimal acceptance cost (see Eq. (1)), whereas for larger values of $k$, we are unaware of a general closed form expression. In Appendix A.1, we provide an information-theoretic upper (and lower) bound, which is tight up to a multiplicative constant of $1 - (1 - 1/k)^k \geq 1 - 1/e$.

While solving OTM in Eq. (4) gives the plan with optimal acceptance probability, to the best of our knowledge, the best-known runtime will be exponential in $k$, which can be prohibitive when either the vocabulary size $|\Omega|$ or the number of draft tokens $k$ is large.[10] In the next section, we will present a selection algorithm that can be efficiently computed and show that it achieves an acceptance probability of at least $(1 - (1 - 1/k)^k)\alpha_k \geq (1 - 1/e)\alpha_k$.

---

[7]The existence of optimal coupling in discrete domain is well-known, *e.g.,* see [27]. When the optimal coupling is not unique, we use $\pi^*$ to denote one of the optimal couplings.

[8]The above generic formulation immediately allows generalization to more complex draft selection strategies, such as sampling $k$ tokens without replacement, or using a different drafting distribution for each draft.

[9]To our best knowledge, the best practical computation bound (through interior-point method) is $O(|\Omega|^{3k})$ [21] and the best theoretical computation bound is $O(|\Omega|^{2.5k})$ [18].

[10]For discrete OT, Sinkhorn algorithm could be used to solve an entropy-regularized version of OT, which has a better computation complexity [6]. However, the computation cost of the algorithm will still have a linear dependence on $|\Omega|^k$, which can be prohibitive.

# 6 Draft selection via $k$-sequential selection

In this section, we present a sequential selection algorithm (K-SEQ), an approximate solution[11] to the optimal transport problem in Eq. (3), which can be efficiently computed in time almost linear in $|\Omega|$ and logarithmic in $k$. The algorithm is presented in Algorithm 2.

---

**Algorithm 2** $k$-sequential selection algorithm (K-SEQ).

---

**Input:** Distributions $p, q$, samples $X_1, \ldots, X_k \sim_{i.i.d.} p$. $\rho \in [1, k]$ : division factor.
  1: Let $\beta_{p,q}(\rho) = \sum_{x \in \Omega} \min(p(x), q(x)/\rho)$ and $p_{\text{acc}} = 1 - (1 - \beta_{p,q}(\rho))^k$. Compute $p^{\text{res}}$ where

$$\forall x \in \Omega, p^{\text{res}}(x) = \frac{q(x) - \min\left\{p(x), \frac{q(x)}{\rho}\right\} \frac{p_{\text{acc}}}{\beta_{p,q}(\rho)}}{1 - p_{\text{acc}}}. \tag{5}$$

  2: **for** $i = 1, 2, \ldots, k$ **do**
  3:     Sample $\eta_i \sim U(0, 1)$.
  4:     **if** $\eta_i \leq \min\left(1, \frac{q(X_i)}{\rho \cdot p(X_i)}\right)$ **then**
  5:         **Return** $Y = X_i$. {*Return the ith draft token.*}
  6:     **end if**
  7: **end for**
  8: **Return** $Y \sim p^{\text{res}}$. {*Sample a corrected token from the residual distribution.*}

---

At a high-level, the algorithm goes over all $k$ draft samples generated from $p$ sequentially, and decides on whether to accept each $X_i$ based on the ratio $q(X_i)/p(X_i)$. The algorithm output the first accepted sample or result from a residual distribution $p^{\text{res}}$ if none of the samples is accepted. To guarantee that the the final returned token is a valid sample from $q$, we choose an appropriate $\rho \in [1, k]$ and accept $X_i$ with probability $\min(1, q(X_i)/(\rho \cdot p(X_i)))$ instead of $\min(1, q(X_i)/(p(X_i)))$ as in Algorithm 1. In Theorem 1, we show that with appropriately chosen $\rho$'s, Algorithm 2 is indeed valid transportation plans from $p^{\otimes k}$ to $q$. Moreover, to find the best transportation plan within the family, we only need to search over a single parameter $\rho$, which reduces the computation cost significantly. We also show that searching over this sub-family of couplings won't decrease the optimal acceptance probability by a multiplicative constant. The performance of Algorithm 2 is stated in Theorem 1.

**Theorem 1.** *Let $\beta_{p,q}(\rho) = \sum_{x \in \Omega} \min\left(p(x), \frac{q(x)}{\rho}\right)$ and $\rho^*$ be the solution to the identity below.*
$$1 - (1 - \beta_{p,q}(\rho))^k = \rho \beta_{p,q}(\rho). \tag{6}$$
*When $\rho \geq \rho^*$, the coupling $\pi_\rho^{\text{K-SEQ}}$ in Algorithm 2 is a valid transport plan from $p^{\otimes k}$ to $q$. When $\rho = \rho^*$, we have*
$$\alpha(\pi_{\rho^*}^{\text{K-SEQ}}) \geq (1 - e^{-1})\alpha_k(p, q).$$
*Moreover, $\rho^*$ can be computed up to accuracy $\delta$ in time $O(|\Omega| \log((k - 1)/\delta))$.*

We provide the proof in Appendix C.1. In Appendix B, using a few canonical examples of distributions, we plot the acceptance probability of K-SEQ and compare it with the optimal acceptance probability $\alpha_k$. It can be shown that K-SEQ could have a strictly worse acceptance probability compared to the OTM solution for certain cases while there also exist non-trivial cases where K-SEQ achieves the optimal acceptance probability.

Concurrent and recent work of [20, 29] has proposed another efficient algorithm for the draft selection phase. To the best of our knowledge, there is no optimality guarantee proved for their proposed algorithm. In Appendix B.3, we present its acceptance probability empirically for the canonical case of Bernoulli distributions, and show that both our proposed algorithms (OTM and K-SEQ) have a higher acceptance probability.

# 7 SpecTr: Application of OTM in autoregressive sampling

In this section, we describe how OTM can be used to speed up auto-regressive sampling, which we refer to as *SpecTr* sampling. Similar to speculative decoding, each iteration of *SpecTr* can be decomposed into three phases (Fig. 2):

---

[11]Note here that the solution still satisfies the constrains in Eq. (3), and hence is a valid transport plan. The term *approximate* here means that the solution is not the exact minimizer of the cost in Eq. (3).

1. **Draft set construction.** Given current context $x^t$, use the draft model sample a set of $K$ draft sequences with length $L$, denoted by $S = \{z^L \sim \mathcal{M}_s(\cdot \mid x^t)\}$. We keep the conditional probabilities $\mathcal{M}_s(y \mid x^t, z^i)$ for all $y \in \Omega, i \leq L$ and $z^L \in S$.
2. **Conditional probability computation.** Compute the conditional probabilities on the next token for the large model $\mathcal{M}_b(y \mid x^t, z^i)$ for all $y \in \Omega, i \leq L$ and $z^L \in S$ in parallel.
3. **Draft selection.** Select first $L'$ of the $L$ tokens and set $x(t+i) = z(i)$ for $i \leq L'$ and some $z \in S$ given the set of draft sequences and the conditional probabilities from both models. Sample a token from a *residual* distribution as a correction to the rejected tokens.

The conditional probability computation step takes $O(1)$ when $|S|$ is not large based on our simplified computations model. We mainly focus on the draft set construction phase and draft selection phase.

---

**Algorithm 3** Draft selection with multiple candidates (DraftSelection).

---

**Input:** Input sequence $x^t$; draft sequence length: $L$; draft sequences $S = \{z_i^L \mid i \leq |S|\}$.
1: Compute a transport plan (using linear programming in Lemma 2 for an optimal solution or Algorithm 2 for a suboptimal solution) from $\mathcal{M}_s(\cdot \mid x^t)^{\otimes|S|}$ to $\mathcal{M}_b(\cdot \mid x^t)$, denoted by $\pi_t$.
2: Get the multi-set of next token-level drafts: $S_z = \{z_i(1)\}_{i \in [|S|]}$ and compute $Y = f_{\pi_t}(S_z)$.
3: **if** $L = 1$ **then**
4:     **if** $Y \in S_z$ **then**
5:         Sample $Y' \sim \mathcal{M}_b(\cdot \mid (x^t, Y))$.
6:         **Return** $(x^t, Y, Y')$. {*Sample an extra token if the last token is accepted.*}
7:     **else**
8:         **Return** $(x^t, Y)$. {*Stop and return the corrected token and previous accepted tokens.*}
9:     **end if**
10: **end if**
11: Let $S_{\text{next}} = \{z^{2:L} \mid z \in S \text{ and } z(1) = Y\}$ be the set that consists of sub-sequences of the candidates that agree with the selected next token.
12: **if** $S_{\text{next}} = \emptyset$ **then**
13:     **Return** $(x^t, Y)$. {*Stop and return the corrected token and previous accepted tokens.*}
14: **else**
15:     **Return** DraftSelection$((x^t, Y), L - 1, S_{\text{next}})$. {*Keep the draft token and proceed to the next time step.*}
16: **end if**

---

**Draft set with *i.i.d.* draft sequences.** Given context $x^t$, a natural way to come up with a set of $K$ drafts is to independently sample $K$ draft sequences from $\mathcal{M}_s(\cdot \mid x^t)$, *i.e.,*

$$z_1^L, z_2^L, \ldots, z_K^L \sim_{i.i.d.} \mathcal{M}_s(\underbrace{\cdot, \cdot, \ldots}_{L \text{ dots}} \mid x^t). \tag{7}$$

The draft set construction method in (7) can be generalized to a prefix-tree based algorithm. However, this generalized version did not perform better in our experiments. We include this construction in Appendix D for completeness.

**Draft selection with multiple candidates.** We present the sequence-level selection algorithm given a set of draft sequences in Algorithm 3. We assume the conditional probabilities on the next token are available given any prefix in the candidate set since they are computed in parallel in the second phase, and won't list them as inputs explicitly in Algorithm 3.

A sample run of the algorithm is presented in Fig. 3. The algorithm proceeds in a recursive fashion. Given prompt $x^t$ and a candidate set $S$ sampled from $\mathcal{M}_s(\cdot \mid x^t)$, the algorithm first computes a token-level draft selection algorithm $f_\pi : \Omega^{|S|} \to \Omega$ which is a transport plan from $\mathcal{M}_s(\cdot \mid x^t)^{\otimes|S|}$ to $\mathcal{M}_b(\cdot \mid x^t)$. Then $f_\pi$ is applied to the set of first tokens of the draft sequences in $S$ to obtained a valid token $Y$ from $\mathcal{M}_b(\cdot \mid x^t)$. If $Y$ is not the last token ($L \geq 2$), we filter out sequences in $S$ whose first token is not $Y$ and denote the remaining sequences as $S_{\text{next}}$ and feed it to the algorithm with context $(x^t, Y)$ and draft length $L - 1$. This goes on until we have $L = 1$ or $S_{\text{next}} = \emptyset$.

In this case when $Y$ is the last token (*i.e.,* $L = 1$) and $Y \in S$, we have the choice to sample an additional token $\mathcal{M}_b(\cdot \mid (x^t, Y))$ since this conditional probability is already computed in the second phase. Due to the property of the token-level selection algorithms and the autoregressive structure of language models, it can be shown that $Y$ is always a valid sample from $\mathcal{M}_b(\cdot \mid x^t)$. Let $L'$ be the number of decoded tokens in one iteration. Note that this is a random variable in the range $[1, L + 1]$.

The formal quality guarantee is stated in Theorem 2. We present the proof in Appendix C.2.

**Theorem 2.** *Assume all drafts in the set $S$ are generated from the small model with input $x^t$, or more precisely, $\forall z \in S$,*

$$\forall i \in [1, L], \qquad z(i) \sim \mathcal{M}_s(\cdot \mid x^t, z^{i-1}). \tag{8}$$

*Let $(x^t, Y^\tau)$ be the output of Algorithm 3 where $\tau$ is the length of the newly decoded tokens, then it satisfies that $Y^{1:\tau}$ is distributed according to $\mathcal{M}_b(\underbrace{\cdot, \cdot, \dots \cdot}_{\tau \text{ dots}} \mid x^t)$. More precisely, For any $\tau_0 \in [1, L+1]$, and any $\tau_0$-length, sequence $o^{\tau_0} = (o(1), \dots, o(\tau_0)) \in \Omega^{\tau_0}$, we have*

$$\Pr\left(Y^{\tau_0} = o^{\tau_0} \mid \tau = \tau_0\right) = \Pi_{i=1}^{\tau_0} \mathcal{M}_b(o(i) \mid x^t, o^{i-1}).$$

## 8 Experiments

We empirically evaluate *SpecTr* and compare it with two methods: (1) the baseline auto-regressive decoding; and (2) speculative decoding with $K = 1$. Note that all three methods effectively generate samples from the same baseline large model, and hence the quality of the two speculative decoding methods is *provably* neutral to that of the large model. Thus, we will only focus on measuring the speedup in our experiments. In the simplified computation model, we made the following assumptions: (1) Decoding time from small models is negligible compared to decoding from the small model; (2) Parallelization along the batch and time axis doesn't increase the time for a serial call to the large model. With these, the theoretical speedup compared to baseline decoding will be the average number of decoded tokens per serial call, which is called *block efficiency* [19], defined below

$$\text{Block efficiency} := \frac{\text{Total number of decoded tokens}}{\text{Number of serial calls to } \mathcal{M}_b}.$$

However, in real deployment of the *SpecTr* algorithm, the actual end-to-end (wall clock) speedup is further impacted by the following aspects. (1) The decoding time for $\mathcal{M}_s$ might not be negligible; (2) Parallelization along the batch and time axis might increase the time for a single call to $\mathcal{M}_b$; (3) Overhead due to the implementation of additional functionalities in *SpecTr* such as the draft selection algorithm and switching between models. These factors will depend on how the algorithm is implemented and optimized. In our experiment, we consider both the block efficiency, and average wall clock speedup with our implementation of *SpecTr*.

We first present the performance of our algorithm and compare it to speculative decoding using state-of-the-art PALM-2 models with prompts from the one-billion language benchmark (LM1B) [3] . In Appendix E, we use a pair of smaller transformer models to break down different affecting factors mentioned above. In Table 1, we use PALM-2-Gecko and PALM-2-Bison as the small model and

Table 1: Experimental results on the LM1B dataset with PALM-2-Gecko as the small model and PALM-2-Bison as the large model. Results are averaged over 1000 test prompts and 3 random seeds.

| Algorithm | $K$ | $L$ | Block efficiency | Relative wall clock speedup (normalized by baseline) |
|---|---|---|---|---|
| Baseline | - | - | 1.0 | 1.0 |
| Speculative | 1 | 4 | 2.4 | 1.67 |
| *SpecTr* | 8 | 4 | **3.1** | **2.08** |
| Speculative | 1 | 8 | 2.9 | 1.56 |
| *SpecTr* | 8 | 8 | **4.0** | **2.13** |

large model, respectively [13, 12]. The wall clock speedup is normalized by the wall clock latency of baseline autoregressive decoding. The time we log include all above mentioned aspects. In the considered parameter configurations, the wall clock speedup increases as $K$ and $L$ increases. As seen from the table, the actual wall clock speedup is smaller than the theoretical speedup of block efficiency, which is consistent with what we expected. Importantly, the benefit from *SpecTr* outweighs these overheads. In particular, when $L = 8$ and $K = 8$, our proposed *SpecTr* algorithm has a speedup of 2.13x, a further 1.37x increase compared to speculative decoding ($K = 1$).

# 9 Acknowledgements

Authors thank Asaf Aharoni, Kwangjun Ahn, Badih Ghazi, Sanjiv Kumar, Teodor Marinov, Michael Riley, and NeurIPS reviewers for helpful comments and discussions.

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

# A Properties of optimal transport cost

## A.1 Information-theoretic upper (and lower) bound of $\alpha_k$.

Below we provide an information-theoretic upper (and lower) bound in Lemma 3, which is tight up to a multiplicative constant of $1 - (1 - 1/k)^k \geq 1 - 1/e$. The proof is presented in Appendix A.3. For the case of $k = 1$, the upper bound matches the optimal acceptance probability.

**Lemma 3.** *For any two distributions $p, q$ and $\forall k \geq 1$, we have*

$$(1 - (1 - 1/k)^k) \cdot \bar{\alpha}_k(p, q) \leq \alpha_k(p, q) \leq \bar{\alpha}_k(p, q),$$

*where*

$$\bar{\alpha}_k(p, q) = \min_{\Omega_0 \subset \Omega} \left\{ \sum_{y \in \Omega_0} \min\left\{ q(y), 1 - (1 - p(y))^k \right\} + \sum_{x^k \in \Omega^k} \min\{\prod_{i=1}^{k} p(x_i), \sum_{y \in x^k \cap \Omega_0^c} q(y)\} \right\}. \tag{9}$$

In Appendix B, we plot $\alpha_k$ as a function of $k$ for a few simple pairs of $(p, q)$'s as illustrative examples. We note that the upper bound in Lemma 3 is tight for examples considered in Appendix B.

## A.2 Proof of Lemma 1

We first prove *monotonicity*. By definition,

$$\alpha_k(p, q) = 1 - \min_{\pi \in \Pi(p^{\otimes k}, q)} \Pr_{X^k, Y \sim \pi}\left(Y \notin S(X^k)\right)$$

$$= \max_{\pi \in \Pi(p^{\otimes k}, q)} \Pr_{X^k, Y \sim \pi}\left(Y \in S(X^k)\right)$$

Moreover, for any $\pi \in \Pi(p^{\otimes k}, q)$, we can construct $\pi' \in \Pi(p^{\otimes k+1}, q)$ by setting

$$\forall x^{k+1} \in \Omega^{k+1}, y \in \Omega, \pi'(x^{k+1}, y) = \pi(x^k, y)p(x(k+1)),$$

*i.e.,* adding and independent sample from $p$ to $X^k$.

Hence we have

$$\alpha_k(p, q) = \max_{\pi \in \Pi(p^{\otimes k}, q)} \Pr_{X^k, Y \sim \pi}\left(Y \in S(X^k)\right)$$

$$= \max_{\pi \in \Pi(p^{\otimes k}, q)} \Pr_{X^{k+1}, Y \sim \pi'}\left(Y \in S(X^k)\right)$$

$$\leq \max_{\pi \in \Pi(p^{\otimes k}, q)} \Pr_{X^{k+1}, Y \sim \pi'}\left(Y \in S(X^{k+1})\right)$$

$$\leq \max_{\pi' \in \Pi(p^{\otimes k+1}, q)} \Pr_{X^{k+1}, Y \sim \pi'}\left(Y \in S(X^{k+1})\right)$$

$$= \alpha_{k+1}(p, q).$$

Next we prove *consistency*. We start with the case when $\forall x \in \Omega, q(x)/p(x) < \infty$. To prove this, we will show that Algorithm 2 with $\rho_{\max} = \max_{x \in \Omega} q(x)/p(x)$ statisfies

$$\lim_{k \to \infty} \alpha(\pi_{\rho_{\max}}^{\text{K-SEQ}}) = 1.$$

Since $\alpha(\pi_{\rho_{\max}}^{\text{K-SEQ}}) \leq \alpha_k(p, q)$, the above equation implies $\lim_{k \to \infty} \alpha_k(p, q) = 1$. Notice that by Lemma 4 and Theorem 1, $\pi_{\rho_{\max}}^{\text{K-SEQ}}$ is a valid coupling, and

$$\alpha(\pi_{\rho_{\max}}^{\text{K-SEQ}}) = 1 - (1 - \beta_{p,q}(\rho_{\max}))^k,$$

where $\beta_{p,q}(\rho) = \sum_{x \in \Omega} \min(p(x), \frac{q(x)}{\rho}) \geq 1/\rho_{\max} > 0$. Taking $k \to \infty$ concludes the proof.

For the case when $q(x)/p(x)$ is unbounded, there exists $x \in \Omega$ such that $q(x) > 0$ and $p(x) = 0$. Let

$$p_{\text{off}} = \sum_{x \in \Omega} \mathbb{1}\left\{p(x) = 0\right\} q(x).$$

Let $x_0$ be such that $p(x_0) > 0$. We define $q'$ such that

$$q' = \begin{cases} 0, & \text{if } p(x) = 0, \\ q(x), & \text{if } p(x) > 0 \text{ and } x \neq x_0, \\ q(x) + p_{\text{off}} & \text{if } x = x_0. \end{cases}$$

Then we have $d_{\text{TV}}(q, q') = p_{\text{off}}$, and hence by subadditivity of transport cost,

$$\alpha_k(p, q) \geq \alpha_k(p, q') - p_{\text{off}}.$$

Moreover, we have $\forall x \in \Omega, q'(x)/p(x) < \infty$. Hence

$$\lim_{k \to \infty} \alpha_k(p, q) \geq \lim_{k \to \infty} \alpha_k(p, q') - p_{\text{off}} = 1 - p_{\text{off}} = \sum_{x \in \Omega} \mathbb{1}\{p(x) > 0\}\, q(x).$$

### A.3 Proof of Lemma 3

For the upper bound, it would be enough to show that for any $\pi \in \Pi(p^{\otimes k}, q)$, and any $\Omega_0 \subset \Omega$, we have

$$\Pr\left(Y \in S(X^k)\right) \leq \sum_{y \in \Omega_0} \min\left\{q(y), 1 - (1 - p(y))^k\right\} + \sum_{x^k \in \Omega^k} \min\{\prod_{i=1}^{k} p(x_i), \sum_{y \in S(x^k) \cap \Omega_0^c} q(y)\}.$$

$$
\begin{aligned}
&\Pr\left(Y \in S(X^k)\right) \\
&= \sum_{y \in \Omega} \sum_{x^k \in \Omega^k} \Pr\left(X^k = x^k, Y = y\right) \cdot \mathbb{1}\left\{y \in S(x^k)\right\} \\
&= \sum_{y \in \Omega_0} \sum_{x^k \in \Omega^k} \Pr\left(X^k = x^k, Y = y\right) \cdot \mathbb{1}\left\{y \in S(x^k)\right\} \\
&\quad + \sum_{y \in \Omega_0^c} \sum_{x^k \in \Omega^k} \Pr\left(X^k = x^k, Y = y\right) \cdot \mathbb{1}\left\{y \in S(x^k)\right\} \\
&= \sum_{y \in \Omega_0} \Pr\left(y \in S(x^k), Y = y\right) + \sum_{x^k \in \Omega^k} \sum_{y \in S(x^k) \cap \Omega_0^c} \Pr\left(X^k = x^k, Y = y\right) \\
&\leq \sum_{y \in \Omega_0} \min\{\Pr\left(y \in S(x^k)\right), q(y)\} + \sum_{x^k \in \Omega^k} \min\{\Pr\left(X^k = x^k\right), \sum_{y \in S(x^k) \cap \Omega_0^c} q(y)\} \\
&= \sum_{y \in \Omega_0} \min\{1 - (1 - p(y))^k, q(y)\} + \sum_{x^k \in \Omega^k} \min\{\prod_{i=1}^{k} p(x_i), \sum_{y \in S(x^k) \cap \Omega_0^c} q(y)\}.
\end{aligned}
$$

For the lower bound, we show that K-SEQ achieves an acceptance probability of at least $(1 - (1 - 1/k)^k)\bar{\alpha}_k(p, q)$, see Eq. (11), implying the lower bound guarantee.

## B  Comparison between $\alpha(\pi_{\rho^*}^{\text{K-SEQ}})$ and $\alpha_k$ for simple examples.

We illustrate the acceptance probabilities for our proposed token-level selection algorithms using a few simple examples and plot them in Figures 4 and 5. The analysis for these simple distributions is presented in Appendix B.1 and Appendix B.2.

**Pairs of Bernoulli distributions.** Let Ber($b$) be a Bernoulli distribution with probability $b$ of getting a head. In Figure 4, we plot the acceptance probability comparison between OTM-$k$ and K-SEQ for different Bernoulli distributions $q = \text{Ber}(b)$ as a function of $k$ when $p = \text{Ber}(0.25)$. Note that when $p = q$ ($b = 0.25$), the acceptance probability is always one for both methods. When $p \neq q$, the acceptance probabilities for both methods increase as $k$ increases before they reach one. When $b = 0.1$ or $0.75$, K-SEQ has a worse acceptance probability compared to the OTM-$k$ algorithm. When $b = 1$, the two algorithms have the same performance.

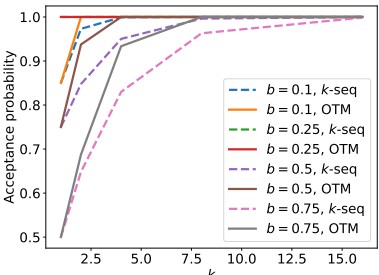
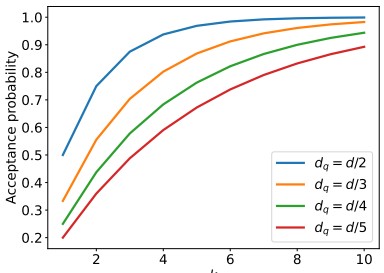

Figure 4: Acceptance probability comparison between OTM-$k$ and K-SEQ when $p =$ Ber(0.25) and $q =$ Ber($b$).

Figure 5: Optimal acceptance probability ($\alpha_k$) as a function of $k$ when $p = U(d)$ for $d = 120$ and $q = U(d_q)$.

**Pairs of uniform distributions.** Let $U(d)$ denote a uniform distribution over $[d]$. In Figure 5, we plot the optimal acceptance probability for different uniform functions $q$ as a function of $k$. For these distributions, it can be shown that K-SEQ achieves the optimal acceptance probability $\alpha_k$. Hence only $\alpha_k$ is plotted. Observe that all acceptance probabilities are monotonically increasing and tend to one when $k \to \infty$, as stated in Lemma 1.

### B.1 Calculations for $\alpha_k$.

In this section, we provide a sketch of optimal acceptance probability calculations for results in Figures 4 and 5.

**Figure 4: Ber($p$) and Ber($q$).** The optimal acceptance probability is

$$\alpha_k(\text{Ber}(p), \text{Ber}(q)) = \min(q, 1 - (1-p)^k) + \min(1-q, 1-p^k). \tag{10}$$

Setting $\Omega_0 = \{0, 1\}$ in Lemma 3 yields the upper bound. For the lower bound observe that since $\Omega = \{0, 1\}$, $\mathbb{1}\left\{y \notin S(x^k)\right\} < 1$ if and only if $x^k$ is $0^k$ or $1^k$. Hence,

$$\alpha_k(\text{Ber}(p), \text{Ber}(q)) = \pi(X^k \notin \{0^k, 1^k\}) + \max_{\pi}\{\pi(Y = 0, X^k = 0^k) + \pi(Y = 1, X^k = 1^k)\}$$

$$= 1 - p^k - (1-p)^k + \max_{\pi}\{\pi(Y = 0, X^k = 0^k) + \pi(Y = 1, X^k = 1^k)\}.$$

Consider the transport plan $\pi$ given by $\pi(1^k, 1) = \min(p^k, q)$, $\pi(1^k, 0) = p^k - \min(p^k, q)$, $\pi(0^k, 0) = \min((1-p)^k, 1-q)$, and $\pi(0^k, 1) = (1-p)^k - \min((1-p)^k, 1-q)$. It can be checked that this is a valid transport plan. To see this matches the upper bound on the optimal cost from Lemma 3, notice that

$$1 - p^k - (1-p)^k + \max_{\pi}\{\pi(Y = 0, X^k = 0^k) + \pi(Y = 1, X^k = 1^k)\}$$

$$= 1 - p^k - (1-p)^k + \min(p^k, q) + \min((1-p)^k, 1-q).$$

If $p^k \leq q$ and $(1-p)^k \leq 1-q$, then the above equation simplifies to 1 and (10) also simplifies to 1. If $p^k > q$ and $(1-p)^k \leq 1-q$, then the above equation simplifies to $1 + q - p^k$ and (10) also simplifies to the same quantity. Similarly, the proof applies for $p^k \leq q$ and $(1-p)^k > 1-q$.

**Figure 5: $p = U(d)$ and $q = U(d/r)$.** The optimal acceptance probability is

$$\alpha_k(U(d), U(d/r)) = 1 - (1 - 1/r)^k.$$

We first prove $\alpha^k(U(d), U(d/r)) \geq 1 - (1 - 1/r)^k$ by a construction. Let $S(X^k)$ be the set of unique symbols in $X^k$. Consider the following transport plan, where $Y$ is drawn uniformly from $S(X^k) \cap [d/r]$ and draws a new uniform sample from $[d/r]$ if $S(X^k) \cap [d/r] = \emptyset$. Observe that since $U(d)$ is uniform over $[d]$, this is a valid transport plan and furthermore,

$$\alpha_k(U(d), U(d/r)) \geq \Pr(S(X)^k \cap [d/r] \neq \emptyset) = 1 - (1 - 1/r)^k.$$

The upper bound follows by setting $\Omega_0 = [d] \setminus [d/r]$ in Lemma 3.

$$\alpha_k(U(d), U(d/r)) \leq \Pr(S(X^k) \cap [d/r] \neq \emptyset) = 1 - (1 - 1/r)^k.$$

## B.2 Acceptance probability of K-SEQ for the example in Figure 5

In this section, we show that for the example in Figure 5, K-SEQ achieves the optimal acceptance accuracy. In this case, $p = U(d)$ and $q = U(d/r)$. Recall that the optimal acceptance probability is

$$\alpha_k(U(d), U(d/r)) = 1 - (1 - 1/r)^k.$$

For $U(d)$ and $U(d/r)$, we have

$$\beta(\rho) = \sum_{x \in [d]} \min\{p(x), q(x)/\rho\} = \frac{1}{\max\{r, \rho\}}.$$

And hence solving $1 - (1 - \beta(\rho))^k = \rho\beta(\rho)$ gives $\rho^* = r(1 - (1 - 1/r)^k)$. And be Theorem 1, we have

$$\alpha(\pi_{\rho^*}^{\text{K-SEQ}}) \geq 1 - (1 - \beta(\rho^*))^k = 1 - (1 - 1/r)^k.$$

And the equality holds since this is an upper bound for any coupling.

## B.3 Comparison to multi-round rejection sampling in [20, 29]

In this section, we compare our proposed draft selection algorithms (OTM and K-SEQ) to the multi-round rejection sampling algorithm (MULTI-ROUND) in concurrent and recent work of [20, 29] (see Algorithm 1 in [29]) using the example of Bernoulli distributions. As Figure 6 demonstrates, both our proposed algorithms outperform their algorithm. The advantage of OTM is demonstrated by the fact it is the optimal algorithm under the validity guarantee of the final accepted token. Our proposed efficient algorithm K-SEQ also outperforms MULTI-ROUND for the considered examples. We leave a systematic comparison of the algorithms as future work.

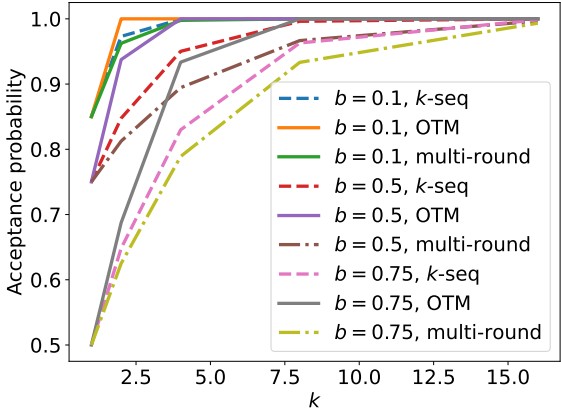

Figure 6: Acceptance probability comparison among OTM-$k$, K-SEQ and MULTI-ROUND when $p = \text{Ber}(0.25)$ and $q = \text{Ber}(b)$. When $b = 0.25$, all three algorithms achieve an acceptance probability of 1, and hence omitted in the plot.

## C Analysis of *SpecTr*

### C.1 Proof of Theorem 1

We start by proving the following lemma on $\rho^*$.

**Lemma 4.** *Let*

$$f(\rho) = 1 - (1 - \beta_{p,q}(\rho))^k - \rho\beta_{p,q}(\rho).$$

*Then we have Let $\rho^*$ be the solution to Eq. (6). Then when $d_{\text{TV}}(p, q) \in (0, 1)$,*

- *$f(\rho)$ is monotone in $\rho$ in $[1, \infty)$;*

- $\rho^* \in \left[1, \min\{k, \max_x \frac{q(x)}{p(x)}\}\right]$.

*Proof.* It would enough to prove the followings: (1) $f(\rho)$ is monotone in $\rho$ in $[1, \infty)$; (2) $f(1) \geq 0$; (3) $f(k) \leq 0$; (4) $f\left(\max_x \frac{q(x)}{p(x)}\right) \leq 0$.

To see (1), since $\beta_{p,q}(\rho)$ is decreasing in $\rho$, so is $1 - (1 - \beta_{p,q}(\rho))^k$. Moreover, $\rho\beta_{p,q}(\rho) = \sum_x \min\{\rho p(x), q(x)\}$, which is non-decreasing in $\rho$. Hence we have $1 - (1 - \beta_{p,q}(\rho))^k - \rho\beta_{p,q}(\rho)$ is decreasing.

To see (2), note that when $\rho = 1$, $\beta_{p,q}(\rho) = 1 - d_{\mathrm{TV}}(p, q)$. Hence we have

$$1 - (1 - \beta_{p,q}(1))^k = 1 - d_{\mathrm{TV}}(p, q)^k \geq 1 - d_{\mathrm{TV}}(p, q).$$

When $\rho = k$, (3) holds since for $x \in [0, 1]$, we have $1 - (1 - x)^k \leq kx$. Moreover, when $\rho = \max_x \frac{q(x)}{p(x)} > 1$, we have $\beta_{p,q}(\rho) = 1/\rho$ and (4) holds since

$$1 - (1 - \beta_{p,q}(\rho))^k = 1 - (1 - 1/\rho)^k < 1 = \rho \cdot 1/\rho.$$

□

Next we prove Theorem 1, we will break the proof into four parts: (1) computation efficiency; (2) $\pi_\rho^{\text{K-SEQ}}$ is a valid transport plan; (3) acceptance probability; (4) optimality guarantee of $\pi_{\rho^*}^{\text{K-SEQ}}$.

**Computation efficiency.** Note that Lemma 4 immediately implies that $\rho^*$ can be computed up to arbitrary accuracy $\delta$ in time $O(|\Omega| \log((k-1)/\delta))$ using binary search over $[1, k]$.

**Valid transport plan.** We next prove that $\pi_\rho^{\text{K-SEQ}}$ is a valid transport plan when $\rho \geq \rho^*$. By Lemma 4, when $\rho \geq \rho^*$, we have $1 - (1 - \beta_{p,q}(\rho))^k \geq \rho\beta_{p,q}(\rho)$. Recall that $p_{\mathrm{acc}} = 1 - (1 - \beta_{p,q}(\rho))^k$, and

$$\forall x \in \Omega, p^{\text{res}}(x) = \frac{q(x) - \min\left\{p(x), \frac{q(x)}{\rho}\right\} \frac{p_{\mathrm{acc}}}{\beta_{p,q}(\rho)}}{1 - p_{\mathrm{acc}}}.$$

$\forall x \in \Omega$, we have

$$\min\left\{p(x), \frac{q(x)}{\rho}\right\} \frac{p_{\mathrm{acc}}}{\beta_{p,q}(\rho)} \leq \frac{1 - (1 - \beta_{p,q}(\rho))^k}{\rho\beta_{p,q}(\rho)} q(x) \leq q(x),$$

this implies $p^{\text{res}}(x) \geq 0$ for all $x \in \Omega$. Moreover,

$$\sum_{x \in \Omega} p^{\text{res}}(x) = \sum_{x \in \Omega} \frac{q(x) - \min\left\{p(x), \frac{q(x)}{\rho}\right\} \frac{p_{\mathrm{acc}}}{\beta_{p,q}(\rho)}}{1 - p_{\mathrm{acc}}} = 1.$$

Hence $p^{\text{res}}$ is a valid distribution. It remains to show that the marginal of $Y$ is $q$. We first compute the probability of the output $Y = x$. Note that probability that $Y = X_1$ is

$$p(X_1) \min\left(1, \frac{q(X_1)}{\rho p(X_1)}\right) = \min\left(p(X_1), \frac{q(X_1)}{\rho}\right).$$

Hence

$$\Pr(Y = X_1 = x) = \min\left(p(x), \frac{q(x)}{\rho}\right).$$

Therefore,

$$\Pr(Y = X_1) = \sum_x \min\left(p(x), \frac{q(x)}{\rho}\right) = \beta(\rho).$$

Similarly, probability that

$$\Pr(Y = X_2 = x) = \Pr(Y \neq X_1)\Pr(Y = X_2 | Y \neq X_1) = (1 - \beta_{p,q}(\rho)) \min\left(p(x), \frac{q(x)}{\rho}\right).$$

Hence,

$$\Pr\left(Y = x, \text{one of } X^k \text{ is accepted}\right)$$

$$= \sum_{i=0}^{k-1} \Pr\left(X_1, \ldots, X_i \text{ are rejected}, X_{i+1} \text{ is accepted, and } X_{i+1} = x\right)$$

$$= \sum_{i=0}^{k-1} (1 - \beta_{p,q}(\rho))^i \cdot p(x) \cdot \min\left\{1, \frac{q(x)}{p(x)\rho}\right\}$$

$$= \min\left\{p(x), \frac{q(x)}{\rho}\right\} \cdot \sum_{i=0}^{k-1} (1 - \beta_{p,q}(\rho))^i$$

$$= \min\left\{p(x), \frac{q(x)}{\rho}\right\} \frac{1 - (1 - \beta_{p,q}(\rho))^k}{\beta_{p,q}(\rho)}$$

Summing over all symbols $x$ yields

$$\Pr(\text{one of } X^k \text{ is accepted}) = \sum_x \min\left\{p(x), \frac{q(x)}{\rho}\right\} \frac{1 - (1 - \beta_{p,q}(\rho))^k}{\beta_{p,q}(\rho)} = 1 - (1 - \beta_{p,q}(\rho))^k.$$

Hence we have

$$\Pr\left(Y = x\right) = \Pr\left(Y = x, \text{one of } X^k \text{ is accepted}\right) + \left(1 - \Pr\left(\text{one of } X^k \text{ is accepted}\right)\right) p^{\text{res}}(x)$$

$$= \min\left\{p(x), \frac{q(x)}{\rho}\right\} \frac{1 - (1 - \beta_{p,q}(\rho))^k}{\beta_{p,q}(\rho)}$$

$$+ \left(1 - (1 - \beta_{p,q}(\rho))^k\right) \frac{q(x) - \min\left\{p(x), \frac{q(x)}{\rho}\right\} \frac{1 - (1 - \beta_{p,q}(\rho))^k}{\beta_{p,q}(\rho)}}{1 - (1 - \beta_{p,q}(\rho))^k}$$

$$= q(x).$$

**Acceptance probability.** The acceptance probability holds since

$$\alpha(\pi_\rho^{\text{K-SEQ}}) \geq \Pr(\text{one of } X^k \text{ is accepted}) = 1 - (1 - \beta_{p,q}(\rho))^k.$$

**Optimality guarantee of $\pi_{\rho^*}^{\text{K-SEQ}}$.** It can be seen that $\beta(\rho)$ is decreasing in $\rho$, and so is $1 - (1 - \beta_{p,q}(\rho))^k$. Hence we have

$$\alpha(\pi_{\rho^*}^{\text{K-SEQ}}) \geq 1 - (1 - \beta_{p,q}(\rho^*))^k \geq 1 - (1 - \beta_{p,q}(k))^k = c_k(p, q) \cdot \sum_x \min\{kp(x), q(x)\},$$

where

$$c_k(p, q) = \frac{1 - (1 - \beta_{p,q}(k))^k}{k\beta_{p,q}(k)} \in [1 - (1 - 1/k)^k, 1).$$

The statement holds since $f(x) = \frac{1 - (1-x)^k}{kx}$ in monotonically decreasing when $x \in (0, 1/k]$ and $f(1/k) = 1 - (1 - 1/k)^k$, $\lim_{x \to 0^+} f(x) = 1$.

Moreover, $\forall x \geq 0, kx \geq 1 - (1 - x)^k$. Hence we have

$$\alpha(\pi_{\rho^*}^{\text{K-SEQ}}) \geq \left(1 - (1 - 1/k)^k\right) \cdot \sum_x \min\{kp(x), q(x)\}$$

$$\geq \left(1 - (1 - 1/k)^k\right) \sum_x \min\{1 - (1 - p(x))^k, q(x)\} \tag{11}$$

$$\geq \left(1 - (1 - 1/k)^k\right) \alpha_k(p, q), nonumber \tag{12}$$

where the last inequality is due to the upper bound in Lemma 3 with $\Omega_0 = \Omega$.

## C.2 Proof of Theorem 2

We prove the theorem via induction. When $L = 1$, $\tau \in \{1, 2\}$. Let $k = |S|$. Since for the first step, $f_\pi$ in Algorithm 3 is a valid transport plan from $\mathcal{M}_s(\cdot \mid x^t)^{\otimes k}$ to $\mathcal{M}_b(\cdot \mid x^t)$. We have $Y_1 \sim \mathcal{M}_b(\cdot \mid x^t)$, which completes the proof when $\tau = 1$. When $\tau = 2$, we have $Y_2 \sim \mathcal{M}_b(\cdot \mid x^t, Y_1)$ as stated in Step 5 of Algorithm 3. Hence the statement holds.

Suppose the theorem holds for $L = \ell \geq 1$, we next prove that it holds for $L = \ell + 1$. Let $Y_{x^t}^\tau$ be the output sequence given context $x^t$. When $\tau = 1$, since for the first step, $f_\pi$ in Algorithm 3 is a valid transport plan from $\mathcal{M}_s(\cdot \mid x^t)^{\otimes k}$ to $\mathcal{M}_b(\cdot \mid x^t)$, we have $Y_1 \sim \mathcal{M}_b(\cdot \mid x^t)$. When $\tau > 1$, $S_{\text{next}} \neq \emptyset$ and by the assumption in Eq. (8), $S_{\text{next}}$ contains $k' = |S_{\text{next}}|$ drafts from $\mathcal{M}_s(\cdot \mid (x^t, Y_1))$ with length $\ell$. Let $Y_{x^t, Y_1}^{\tau'}$ be the output sequence given context $(x^t, Y_1)$, by the induction assumption, we have for any $\tau_0 \in [1, \ell + 1]$, and any $\tau_0$-length, sequence $o^{\tau_0} = (o(1), \ldots, o(\tau_0)) \in \Omega^{\tau_0}$, we have

$$\Pr\left(Y_{x^t, Y_1}^{\tau'} = o^{\tau_0} \mid \tau' = \tau_0\right) = \Pi_{i=1}^{\tau_0} \mathcal{M}_b(o(i) \mid x^t, Y_1, o^{i-1}).$$

Note that in this case $\tau = \tau' + 1$, and for any $(\tau_0 + 1)$-length sequence $o^{\tau_0+1} = (o(1), \ldots, o(\tau_0), o(\tau_0 + 1)) \in \Omega^{\tau_0+1}$, we have

$$\Pr\left(Y_{x^t}^\tau = o^{\tau_0+1} \mid \tau = \tau_0 + 1\right) = \Pr\left(Y_1 = o_1\right) \cdot \Pr\left(Y_{x^t, o_1}^{\tau'} = o^{\tau_0} \mid \tau' = \tau_0\right)$$
$$= \mathcal{M}_b(o(1) \mid x^t) \cdot \Pi_{i=1}^{\tau_0} \mathcal{M}_b(o(i + 1) \mid x^t, o^i)$$
$$= \Pi_{i=1}^{\tau_0+1} \mathcal{M}_b(o(i) \mid x^t, o^{i-1}).$$

Combining the two cases, we complete the proof.

## D  Candidate set construction via a prefix-tree

As discussed in Section 1, the size of the draft set $S$ is constrained by the number of parallel computations that can be supported in the hardware. Hence it is important to design the draft set carefully to allow for a longer sequence of accepted candidate sets. In addition to the *i.i.d.* draft set selection approach listed in Section 7, we present an algorithm that samples a draft set that forms the leaves of a prefix tree. Given a draft set size $K$, the algorithm can be specified by a sequence of parameter $(k_1, k_2, \ldots, k_L)$ satisfying $\prod_{i=1}^{L} k_i = K$.

The algorithm starts with a root node with sequence $x^{1:t}$ and forms a prefix tree of depth $L$. At depth $i \in [1 : L - 1]$, each node is expanded by a factor of $k_{i+1}$ and each of its children will contain a sequence that satisfies: (1) Its prefix agrees with the sequence in the parent node; (2) The next token is sampled from the conditional probability given the prefix in small model. These child nodes will be at depth $i + 1$ and the process goes until it hits depth $L$. We give a detailed description of the algorithm in Algorithm 4.

---

**Algorithm 4** Draft set selection via prefix-tree.

---

**Input:** Input sequence $x^t$; expansion factors at each level: $(k_1, k_2, \ldots, k_L)$.
1: $S_0 = \{x^t\}$.
2: **for** $i = 0, 1, 2, \ldots, L - 1$ **do**
3:     $S_{i+1} = \emptyset$.
4:     **for** all seq $\in S_i$ **do**
5:         Sample $k_{i+1}$ *i.i.d.* tokens $X_1, X_2, \ldots, X_{k_{i+1}}$ from $\mathcal{M}_b(\cdot \mid \text{seq})$.
6:         $S_{i+1} = S_{i+1} \cup \{(\text{seq}, X_i), i = 1, 2, \ldots k_{i+1}\}$.
7:     **end for**
8: **end for**
9: **Return** $S_L$.

---

## E  Additional experiments

In this section, we perform a detailed investigation of different factors that affect the speed of *SpecTr* with smaller transformer models. We train decoder-only transformer models on the LM1B

Table 2: Average latency with parallelization along the time axis and batch axis. We report average latency with standard deviation over 1,000 runs using a $97M$ transformer relative to length = 4 and batch = 1 on GPU.

| Relative latency | batch = 1 | batch = 2 | batch = 4 | batch = 8 |
|---|---|---|---|---|
| length = 4 | $1.00 \pm 0.16$ | $1.01 \pm 0.15$ | $1.06 \pm 0.10$ | $1.10 \pm 0.16$ |
| length = 8 | $1.01 \pm 0.18$ | $1.09 \pm 0.25$ | $1.10 \pm 0.09$ | $1.42 \pm 0.4$ |

Table 3: Average latency with parallelization along the time axis and batch axis. We report average latency with standard deviation over 1,000 runs using a $6M$ transformer relative to the $97M$ transformer with length = 4 and batch = 1 on GPU.

| Relative latency | batch = 1 | batch = 2 | batch = 4 | batch = 8 |
|---|---|---|---|---|
| length = 4 | $0.18 \pm 0.02$ | $0.19 \pm 0.04$ | $0.18 \pm 0.09$ | $0.20 \pm 0.13$ |
| length = 8 | $0.17 \pm 0.04$ | $0.19 \pm 0.05$ | $0.16 \pm 0.02$ | $0.18 \pm 0.04$ |

dataset based on the example provided in the FLAX library [15]. For the draft model, we use transformer models with $2M$, $6M$ and $20M$ parameters, and for the large model we use a $97M$ parameter transformer model.

We first provide a verification of the computational model introduced in Section 1 by reporting the latencies of using the large model to compute the probabilistic distributions with parallelization over time and batch axes. As shown in Table 2, the latency stays roughly constant in these setting.

Similar to Table 2, we report relative latency when parallelizing across the time and batch axes using the small $6M$ draft model in Table 3. In Table 3, the reported relative latencies are relative to the large $97M$ model to get a sense of the relative cost of sampling multiple drafts with the small model compared to the large model.

To see how the size of size of the draft model will affect the block efficiency, we also include results for varying draft model sizes with the same $97M$ large model for LM1B in Table 4. These draft models were produced by either halving ($2M$) or doubling ($20M$) the original $6M$ draft model's number of layers, embedding dimension, MLP dimension, and number of attention heads. As expected, the larger draft models improve all speculative methods' block efficiency with SpecTr maintaining the best performance across all draft model sizes.

Table 4: Experimental results on the LM1B dataset with varying draft model sizes and the $97M$ transformer as the large model. All results are over 1000 test prompts averaged over three different random seeds and sampling temperature of 1.0 for both the draft and large models.

| Draft model | Algorithm | $K$ | $L$ | Block efficiency |
|---|---|---|---|---|
| $2M$ Transformer | Baseline | - | - | 1.00 |
| | Speculative | 1 | 4 | $1.86 \pm 0.02$ |
| | *SpecTr* | 2 | 4 | $2.07 \pm 0.01$ |
| | *SpecTr* | 4 | 4 | $2.32 \pm 0.00$ |
| | *SpecTr* | 8 | 4 | $\mathbf{2.56 \pm 0.01}$ |
| | Speculative | 1 | 8 | $1.91 \pm 0.01$ |
| | *SpecTr* | 2 | 8 | $2.15 \pm 0.01$ |
| | *SpecTr* | 4 | 8 | $2.41 \pm 0.00$ |
| | *SpecTr* | 8 | 8 | $\mathbf{2.68 \pm 0.01}$ |
| $6M$ Transformer | Baseline | - | - | 1.00 |
| | Speculative | 1 | 4 | $2.21 \pm 0.01$ |
| | *SpecTr* | 2 | 4 | $2.43 \pm 0.01$ |
| | *SpecTr* | 4 | 4 | $2.74 \pm 0.01$ |
| | *SpecTr* | 8 | 4 | $\mathbf{2.99 \pm 0.02}$ |
| | Speculative | 1 | 8 | $2.33 \pm 0.01$ |
| | *SpecTr* | 2 | 8 | $2.61 \pm 0.02$ |
| | *SpecTr* | 4 | 8 | $2.96 \pm 0.03$ |
| | *SpecTr* | 8 | 8 | $\mathbf{3.27 \pm 0.02}$ |
| $20M$ Transformer | Baseline | - | - | 1.00 |
| | Speculative | 1 | 4 | $2.71 \pm 0.01$ |
| | *SpecTr* | 2 | 4 | $2.96 \pm 0.00$ |
| | *SpecTr* | 4 | 4 | $3.28 \pm 0.02$ |
| | *SpecTr* | 8 | 4 | $\mathbf{3.49 \pm 0.03}$ |
| | Speculative | 1 | 8 | $3.12 \pm 0.02$ |
| | *SpecTr* | 2 | 8 | $3.48 \pm 0.04$ |
| | *SpecTr* | 4 | 8 | $3.85 \pm 0.05$ |
| | *SpecTr* | 8 | 8 | $\mathbf{4.15 \pm 0.04}$ |

