# SpecTr: Fast Speculative Decoding via Optimal Transport

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

 prefix and sample the next sequence using speculative decoding iteratively. For completeness, we provide the full algorithm in Appendix A. The crux of the above three steps is draft selection, which given a draft sequence and the conditional probabilities from both models, selects a valid sequence such that the output has same distribution as that of the large model. In speculative decoding, this is achieved via recursively applying a token-level maximal coupling algorithm, which is provided in Algorithm 1. Note that for the draft selection, Algorithm 1 is applied where $p$ is the conditional distribution of the draft model $\mathcal{M}_s(\cdot \mid x^t)$ and $q$ is the conditional distribution of the large model $\mathcal{M}_b(\cdot \mid x^t)$ (which may be further conditioned on the context of the language model).

---

**Algorithm 1** Token-level maximal coupling

---

**Input:** Distributions $p, q$, Draft sample $X \sim_{i.i.d.} p$.
  1: Compute $p^{\text{res}}$ where $\forall x \in \mathcal{X}, p^{\text{res}}(x) = \frac{q(x) - \min\{p(x), q(x)\}}{1 - \sum_{x'} \min\{p(x'), q(x')\}}$.
  2: Set $Y = \perp$.
  3: Sample $\eta \sim U(0, 1)$.
  4: **if** $\eta \leq \min\left(1, \frac{q(X)}{p(X)}\right)$ **then**
  5:     $Y = X, accept = \texttt{True}$
  6: **end if**
  7: **Return** $Y \sim p^{\text{res}}$, $accept = \texttt{False}$.

---

Algorithm 1 returns a random variable $Y$ which either is the accepted input $X$ ($accept = \texttt{True}$) or a sample from the residual distribution $p^{\text{res}}$ ($accept = \texttt{False}$), which is defined in Step 1 of Algorithm 1. The algorithm is recursively applied as long as the draft tokens are accepted ($accept = \texttt{True}$) to select the first $L' \leq L$ tokens from the draft model. Previous works showed that if $X \sim p$, then $Y \sim q$ [15, 4]. In the case of the draft selection this means that the output of the algorithm is distributed according to $\mathcal{M}_b(\cdot \mid x^t)$, which is exactly the desired outcome. Furthermore

$$\Pr(Y = X) = \sum_{x \in V} \min(p(x), q(x)) = 1 - d_{\text{TV}}(p, q),$$

where $d_{\text{TV}}$ is the total variation distance between $p$ and $q$. Since $Y$ is distributed according to $q$, it is a valid sample from the large model. Secondly, the more similar $p$ and $q$ are, the higher the chance of $\Pr(Y = X)$, and fewer the number of serial calls to the larger model. In the ideal case, if $p = q$, then $\Pr(Y = X) = 1$, i.e., the draft token is always accepted, and when used for speculative decoding we have $L' = L$. In such a case, based on our computational model (Section 1), assuming the draft model is very fast compared to the large model, the speedup is $L$ times.

# 3 Our contributions

From a theoretical viewpoint, the speculative decoding algorithm raises multiple questions.

- What is the relationship between speculative decoding and the broader literature of sampling in statistics?

- Is speculative decoding optimal in an information-theoretic sense?

- Speculative decoding uses parallelization along time to speed up decoding, would it be possible to use parallelization along batch (number of drafts) to further improve decoding speed?

We provide answers to all the above questions in this work. We first relate the problem of speculative decoding to the broader and well-studied discrete optimal transport theory (Section 4). With this connection, it becomes clear that the token-level draft selection is the optimal solution for optimal transport with indicator cost function and also related to the problem of maximal coupling [7]. Based on the connection to optimal transport, we show that one can further speed up the decoding by parallelizing along the batch axis by using multiple drafts from the draft model (Section 5).

More precisely, we formulate the draft selection problem as an discrete optimal transport problem with membership cost. Discrete optimal transport can be solved with a linear program, but the number of variables is exponential in batch size, which can be prohibitive. To address this, we propose an

approximate solution which achieves a $(1-1/e)$-approximation of the optimal acceptance probability (Section 6).

With this theoretically motivated algorithm and guarantees, we circle back to speeding up decoding and propose a new algorithm called SpecTr and theoretically show that it can be used to derive valid sequences from the large model (Section 7). We then experimentally demonstrate the benefit of our approach on standard datasets (Section 8).

# 4 Token-level draft selection as an optimal transport problem

In this section, we formulate token-level draft as an optimal transport problem, where a cost function is associated with whether a draft token is accepted. To simplify notations, we assume the data comes from a discrete domain, but this can be easily generalized.

**Definition 1** (Coupling). For two probability distributions $P$ over $\mathcal{X}$ and $Q$ over $\mathcal{Y}$, we say a joint distribution $\pi$ supported over $\mathcal{X} \times \mathcal{Y}$ is a coupling between $P$ and $Q$ if

$$\forall y \in \mathcal{Y}, \ \sum_{x \in \mathcal{X}} \pi(x, y) = Q(y),$$

$$\forall x \in \mathcal{X}, \ \sum_{y \in \mathcal{Y}} \pi(x, y) = P(x).$$

We use $\Pi(P, Q)$ to denote the set of all possible couplings between $P$ and $Q$.

When it is clear from context, we will overload notation and refer to the probabilistic mapping $f_\pi : \mathcal{X} \to \mathcal{Y}$ introduced by the conditional probability $\pi(y \mid x) := \pi(x, y)/P(x)$ as a coupling, which is also referred to the transport plan from $P$ to $Q$ [22].

**Definition 2** (Optimal Transport (OT) [22]). For a cost function $c : \mathcal{X} \times \mathcal{Y} \to \mathbb{R}_+$, the *transportation cost* of a coupling is defined as:

$$C(\pi) = \mathbb{E}_{X,Y \sim \pi} \left[ c(X, Y) \right].$$

The *optimal transport plan* is the coupling $\pi \in \Pi(P, Q)$ that minimizes the transportation cost.

With these definitions in place, we can see that with $\mathcal{X} = \mathcal{Y} = \Omega$, which is the alphabet of the tokens, we recover the speculative decoding with the cost function of *indicator cost*, which captures the resampling cost, defined below:

$$\forall x \in \Omega, y \in \Omega, \qquad c(x, y) = \mathbb{1}\{y \neq x\}.$$

The transportation cost of the coupling will be

$$C(\pi) = \mathbb{E}_{X,Y \sim \pi} \left[ \mathbb{1}\{Y \neq X\} \right] = \mathbb{P}_{X,Y \sim \pi}(Y \neq X).$$

This optimal transport with this specific cost function is also called maximal coupling [7], and the optimal cost is known to be

$$\min_{\pi:\Pi} \mathbb{P}_{X,Y \sim \pi}(Y \neq X) = \sum_{x \in \Omega} \min(P(x), Q(x)). \tag{1}$$

Moreover, it can be shown that Algorithm 1 is equivalent to the maximal coupling between $p$ and $q$, and hence it achieves the optimal cost [7].

# 5 Optimal transport with multiple draft tokens

In this section, we generalize speculative decoding to allow for multiple drafts. More formally, let $\mathcal{X} = \Omega^k$ for some $k \in \mathbb{N}_+$, which is the set of $k$ draft tokens from $\Omega$ and $\mathcal{Y} = \Omega$, which is the space of the final sampled token from the desired distribution. To characterize the resampling cost, we use the cost function of *membership cost*, defined below:

$$\forall x \in \Omega^k, y \in \Omega, \qquad c(x, y) = \mathbb{1}\{y \notin S(x)\},$$

where $S(x) = \{o \in \Omega \mid o \text{ appears in } x\}$ denotes the set of distinct elements in $x$. When $k = 1$, this recovers the indicator cost mentioned above. The transportation cost of the coupling will be

$$C(\pi) = \mathbb{E}_{X,Y \sim \pi} \left[ \mathbb{1} \left\{ Y \notin S(X) \right\} \right] = \mathbb{P}_{X,Y \sim \pi}(Y \notin S(X)). \tag{2}$$

We will also refer to the above cost $C(\pi)$ as the *rejection probability* due to its probabilistic interpretation. And similarly, $\alpha(\pi) := 1 - C(\pi) = (Y \in S(X))$ will be the *acceptance probability*.

From now on we will use membership cost as the default cost function and refer to the optimal transport solution as *optimal transport with membership cost* (OTM). We use $\pi^*$ to denote the coupling that minimizes this cost $\pi^* = \arg\min_{\pi \in \Pi(P,Q)} C(\pi);$[2] and the cost $C(\pi^*)$ is referred to as the *optimal transport cost* between $P$ and $Q$. We use $\alpha(P, Q) = 1 - C(\pi^*)$ to denote the corresponding optimal acceptance probability.

**Draft selection with *i.i.d.* draft tokens.**   In this paper, we will mainly focus on the case when the draft tokens are *i.i.d.* samples from a base distribution. Let $p, q$ be supported over $\Omega$ and the goal is to obtain one valid token from $q$ given $k$ *i.i.d.* samples from $p$. This applies to the practical scenario where there exists a computationally efficient model, from which we can sample multiple independent draft tokens efficiently. We set $P = p^{\otimes k}$, a product distribution whose marginals are all $p$, and $Q = q$. And the OT problem we want to solve is the following:

$$\min C(\pi) \ \ s.t. \ \ \pi \in \Pi(p^{\otimes k}, q). \tag{3}$$

In this case, we overload notation and denote the *optimal acceptance probability* as $\alpha_k(p, q) := \alpha(p^{\otimes k}, q) = 1 - C(\pi^*)$. To better understand the quantity, below we state a few properties of $\alpha_k$.

**Lemma 1.**  *(Appendix B.1)  The optimal acceptance probability statisfies the following properties.*

- ***Monotonicity.** For any $p, q$ and $k \geq 1$, $\alpha_k(p, q) \leq \alpha_{k+1}(p, q)$.*

- ***Consistency.** If $q(x)/p(x)$ is bounded $\forall x \in \Omega$, we have*

$$\lim_{k \to \infty} \alpha_k(p, q) = 1.$$

*Else,*

$$\lim_{k \to \infty} \alpha_k(p, q) = \sum_{x \in \Omega} \mathbb{1} \left\{ p(x) > 0 \right\} q(x).$$

With the above result, it is clear that increasing $k$ might increase the acceptance probability, particularly when the draft model satisfies $p(x) > 0$ for all $x \in \Omega$. We now focus on computing the optimal transport scheme and the optimal acceptance probability. Optimal transport in discrete domain has been studied extensively [14, 16, 11], and it is shown that the optimal transport problem is equivalent to the following linear programming problem:

$$\min \sum_{x \in \Omega^k} \sum_{y \in \Omega} \pi(x, y) \mathbb{1} \left\{ y \notin S(x) \right\} \tag{4}$$

$$s.t. \quad \forall y \in \Omega, \ \sum_x \pi(x, y) = Q(y)$$

$$\forall x \in \Omega^k, \ \sum_y \pi(x, y) = P(x)$$

$$\forall x \in \Omega^k, y \in \Omega, \ \pi(x, y) \geq 0.$$

Linear programming can be solved in time polynomial in the number of variables and constraints [6, 16]. Linear program in (4) has $|\Omega|^{k+1}$ variables and $|\Omega|^k + |\Omega|$ equality constraints.

**Lemma 2.**  *Given $p, q$ over $\Omega$, there exists an algorithm that computes a solution to Eq. (3) in time $O(|\Omega|^{O(k)})$.*

---

[2]The existence of optimal coupling in discrete domain is well-known, *e.g.,* see [22]. When the optimal coupling is not unique, we use $\pi^*$ to denote one of the optimal couplings.

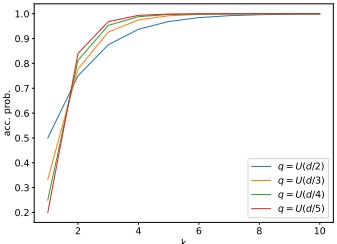

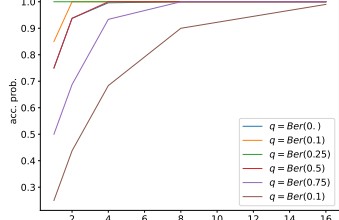

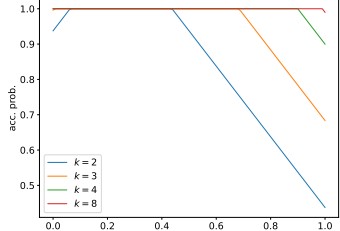

Figure 1: Optimal acc. prob. as a function of $k$ when $p = U(d)$ for $d = 120$.

Figure 2: Optimal acc. prob. as a function of $k$ when $p = \text{Ber}(0.25)$.

Figure 3: Optimal acc. prob. as a function of $q$ when $p = \text{Ber}(0.25)$.

We refer to the optimal coupling obtained above as OTM-$k$ and denote it as $\pi^{\text{OTM}-k}$. For the case of $k = 1$, we have a closed form expression for the optimal acceptance cost (see Eq. (1)), whereas for larger values of $k$, we don't have a general closed form expression.

We illustrate for few simple cases and plot them in Figures 1, 2, 3 and provide analysis for these simple distributions in Appendix B.2. Let $U(d)$ denote a uniform distribution over $[d]$. In Figure 1, we plot the optimal acceptance probability for different uniform functions $q$ as a function of $k$. Observe that all acceptance probabilities are monotonically increasing and tend to one as $k \to \infty$, however the rate of convergence is vastly different. Furthermore if $\alpha_1(p, q) > \alpha_1(p, q')$, that does not necessarily mean $\alpha_k(p, q) > \alpha_k(p, q')$. In Figure 2, we plot the optimal acceptance probability for different Bernoulli distributions $q$ as a function of $k$ when $p = \text{Ber}(0.25)$. Note that when $p = q$, the acceptance probability is always one (green line), but as we increase / decrease $q$ the acceptance probability decreases. Finally, in Figure 3, we plot the acceptance probability for different values of $k$ as a function of $q$, when $p = \text{Ber}(0.25)$. In this scenario, note that if $k$ is sufficiently large, say 8, then for most values of $q$, the acceptance probability is one, however if $k$ is small, then the acceptance probability depends on how close $p$ and $q$ are. Even though, we don't have a closed form solution for general $k$, we provide an information-theoretic upper bound in the next theorem. For the case of $k = 1$, this upper bound matches the optimal acceptance probability of previous results. We also note that this bound is tight for all of the above examples.

**Theorem 1** (Appendix B.3). *For any two distributions $p, q$ and $\forall k \geq 1$, we have*

$$\alpha_k(p, q) \leq \min_{\Omega_0 \subset \Omega} \left\{ \sum_{y \in \Omega_0} \min\left\{q(y), 1 - (1 - p(y))^k\right\} + \sum_{x^k \in \Omega^k} \min\{\prod_{i=1}^{k} p(x_i), \sum_{y \in x^k \cap \Omega_0^c} q(y)\} \right\}.$$

While this solution gives the optimal transportation cost, if we aim to use generic linear program solver to solve (4), to the best of our knowledge, the best-known runtime will be exponential in $k$, which can be prohibitive when either the vocabulary size $\Omega$ or the number of draft tokens $k$ is large. In the next section, we will present an approximate solution to the OTM problem and show that for any pair of distributions, it gives a $(1 - 1/e)$ approximation to the *optimal acceptance probability* $\alpha_k$.

## 6 Approximate OTM via $k$-sequential selection

In this section, we present sequential selection algorithm (K-SEQ), an approximate solution to the optimal transport problem in Eq. (3), which can be efficiently computed in time almost linear in $|\Omega|$ and logarithmic in $k$. The algorithm is presented in Algorithm 2.

At a high-level, the algorithm goes over all $k$ samples $X_1, \ldots, X_k$ generated from $p$ sequentially, and decides on whether to accept each $X_i$ based on the ratio $q(X_i)/p(X_i)$. The algorithm output the first accepted sample or result from a residual distribution $p^{\text{res}}$ if none of the samples is accepted. To control the probability of accepting an $x \in \Omega$ with probability larger than $q(x)$. We choose an appropriate $\gamma \in [1, k]$ and accept $X_i$ with probability $\min(1, q(X_i)/(\gamma \cdot p(X_i)))$ instead of $\min(1, q(X_i)/(p(X_i)))$ as in the single-draft case. Further, notice that Algorithm 2 recovers Algorithm 1 when $\gamma = k = 1$.

---
**Algorithm 2** $k$-sequential selection algorithm (K-SEQ).

---
**Input:** Distributions $p, q$, samples $X_1, \ldots, X_k \sim_{i.i.d.} p$. $\gamma \in [1, k]$ : division factor.

1: Let $\beta_{p,q}(\gamma) = \sum_{x \in \Omega} \min(p(x), q(x)/\gamma)$ and $p_{\mathrm{acc}} = 1 - (1 - \beta_{p,q}(\gamma))^k$. Compute $p^{\mathrm{res}}$ where

$$\forall x \in \Omega, p^{\mathrm{res}}(x) = \frac{q(x) - \min\left\{p(x), \frac{q(x)}{\gamma}\right\} \frac{p_{\mathrm{acc}}}{\beta_{p,q}(\gamma)}}{1 - p_{\mathrm{acc}}}. \tag{5}$$

2: **for** $i = 1, 2, \ldots, k$ **do**
3:     Sample $\eta_i \sim U(0, 1)$.
4:     **if** $\eta_i \leq \min\left(1, \frac{q(X_i)}{\gamma \cdot p(X_i)}\right)$ **then**
5:         $Y = X_i$.
6:         **Return** $Y = X_i$.
7:     **end if**
8: **end for**
9: **Return** $Y \sim p^{\mathrm{res}}$.

---

In Theorem 2, we show that family of joint distributions induced by Algorithm 2 is indeed valid transportation plans from $p^{\otimes k}$ to $q$. Moreover, to find the best transportation plan within the family, we only need to search over a single parameter $\gamma$, which reduces the computation cost significantly. We also show that searching over this sub-family of couplings won't decrease the optimal acceptance probability by a multiplicative constant. The performance of Algorithm 2 is stated in Theorem 2.

**Theorem 2.** *Let* $\beta_{p,q}(\gamma) = \sum_{x \in \Omega} \min(p(x), \frac{q(x)}{\gamma})$ *and* $\gamma^*$ *be the solution to the identity below.*

$$1 - (1 - \beta_{p,q}(\gamma))^k = \gamma \beta_{p,q}(\gamma). \tag{6}$$

*When* $\gamma \geq \gamma^*$, *the coupling* $\pi_\gamma^{\text{K-SEQ}}$ *introduced by Algorithm 2 is a valid transport plan from* $p^{\otimes k}$ *to* $q$ *and*

$$\alpha(\pi_\gamma^{\text{K-SEQ}}) \geq p_{\mathrm{acc}} = 1 - (1 - \beta_{p,q}(\gamma))^k.$$

*And when* $\gamma = \gamma^*$, *we have*

$$\alpha(\pi_{\gamma^*}^{\text{K-SEQ}}) \geq (1 - e^{-1})\alpha_k(p, q).$$

*Moreover,* $\gamma^*$ *can be computed in time* $O(|\Omega| \log k)$.

Due to space constraints, we provide the proof in the appendix. To see why $\gamma^*$ can be computed efficiently, we notice that the function $f(\gamma)$ defined below has a root in $[1, k]$. Moreover it is continuous and monotonically increasing when $\gamma \in [1, k]$:

$$f(\gamma) = 1 - (1 - \beta_{p,q}(\gamma))^k - \gamma \beta_{p,q}(\gamma).$$

Hence the solution to Eq. (6) can be efficiently computed using binary search over the set $[1, k]$.

In fact, although Theorem 2 only guarantees that Algorithm 2 can achieve an acceptance rate at least a $(1 - e^{-1})$ factor of the optimal acceptance rate, empirically we observe that the acceptance probabilities are much closer for certain distributions. For example, for all the examples listed in the previous section, the proposed algorithm is in fact optimal. We list few more comparisons in the appendix.

## 7 SpecTr: Application of OTM in autoregressive sampling

In this section, we describe how OTM can be used to speed up auto-regressive sampling, which we refer to as SpecTr sampling. Similar to speculative decoding, each step of SpecTr can be decomposed into three phases:

1. **Draft set construction.** Given context $x^T$, use the draft model sample a set of draft sequences with length $L$, denoted by $S = \{z^L \sim \mathcal{M}_s(\cdot \mid x^t)\}$. We keep the conditional probabilities $\mathcal{M}_s(y \mid x^t, z^i)$ for all $y \in \Omega, i \leq L$ and $z^L \in S$.
2. **Conditional probability computation.** Compute the conditional probabilities on the next token for the large model $\mathcal{M}_b(y \mid x^t, z^i)$ for all $y \in \Omega, i \leq L$ and $z^L \in S$ in parallel.

3. **Draft selection.** Select first $L'$ of the $L$ tokens and set $x(t + i) = z(i)$ for $i \leq L'$ and some $z \in S$ given the set of draft sequences and the conditional probabilities from both models.

---

**Algorithm 3** Draft selection with multiple candidates (DraftSelection).

---

**Input:** Input sequence $x^t$; candidate length: $L$; a set of candidates $S = \{z_i^L \mid i = 1, \ldots, |S|\}$ with length $L$.

1: Compute a transport plan (using linear programming in Lemma 2 for an exact solution or Algorithm 2 for an approximate solution) from $\mathcal{M}_s(\cdot \mid x^t)^{\otimes |S|}$ to $\mathcal{M}_b(\cdot \mid x^t)$, denoted by $\pi_t$.
2: Get the multi-set of next token-level drafts: $S_z = \{z_i(1)\}_{i \in [|S|]}$ and compute $Y = f_{\pi_t}(S_z)$.
3: **if** $L = 1$ **then**
4:    **if** $Y \in S_z$ **then**
5:       Sample $Y' \sim \mathcal{M}_b(\cdot \mid (x^t, Y))$
6:       **Return** $(x^t, Y, Y')$.
7:    **else**
8:       **Return** $(x^t, Y)$
9:    **end if**
10: **end if**
11: Let $S_{\text{next}} = \{z^{2:L} \mid z \in S \text{ and } z(1) = Y\}$ be the set that consists of sub-sequences of the candidates that agree with the selected next token.
12: **if** $S_{\text{next}} = \emptyset$ **then**
13:    **Return** $(x^t, Y)$
14: **else**
15:    **Return** DraftSelection$((x^t, Y), L - 1, S_{\text{next}})$
16: **end if**

---

**Draft set with *i.i.d.* draft sequences.** Gvien context $x^t$, a natural way to come up with a set of $K$ drafts is to independently sample $K$ draft sequences from the conditional distribution $\mathcal{M}_s(\cdot \mid x^t)$, *i.e.*,

$$z_1^L, z_2^L, \ldots, z_K^L \sim_{i.i.d.} \mathcal{M}_s(\underbrace{\cdot, \cdot, \ldots \cdot}_{L \text{ dots}} \mid x^t).$$
(7)

The draft set construction method in (7) can be generalized to a prefix-tree based algorithm. However, this generalized version did not perform better in experiments. We include this construction in the appendix for completeness

**Draft selection with multiple candidates.** We present the selection algorithm given a set of draft sequences in Algorithm 3. We assume the condition probabilities on the next token is available given any prefix in the candidate set since they are computed parallelly in the second phase, and won't list them as inputs explicitly in Algorithm 3.

A sample run of the algorithm is presented in Fig. 4. The algorithm proceeds in a recursive fashion. Given prompt $x^t$ and a candidate set $S$ sampled from $\mathcal{M}_s(\cdot \mid x^t)$, the algorithm first computes the optimal transport plan $f_\pi : \Omega^{|S|} \to \Omega$ from $\mathcal{M}_s(\cdot \mid x^t)^{\otimes |S|}$ to $\mathcal{M}_b(\cdot \mid x^t)$. Then $f_\pi$ is applied to the first token in each sequence in $S$ to obtained a valid token $Y$ from $\mathcal{M}_b(\cdot \mid x^t)$. If $Y$ is not the last token ($L \geq 2$), we filter out sequences in $S$ whose first token is not $Y$ and denote the remaining sequences as $S_{\text{next}}$ and feed it to the algorithm with context $(x^t, Y)$ and draft length $L - 1$. This goes on until we have $L = 1$ or $S_{\text{next}} = \emptyset$.

In this case when $Y$ is the last token (*i.e.*, $L = 1$) and $Y \in S$, we have the choice to sample an additional token $\mathcal{M}_b(\cdot \mid (x^t, Y))$ since this conditional probability is already computed in the second

| $|S_z| = 6$ | $|S_z| = 3$ | $|S_z| = 2$ | $|S_z| = 1$ |
|---|---|---|---|
| be | liked | by | all |
| be | read | by | four |
| be | liked | for | its |
| not | be | liked | by |
| not | get | good | reviews |
| receive | one | good | review |

*This paper will*

Figure 4: An example of draft selection in SpecTr with $L = 4$ and $K = 6$. Draft selection algorithm has input of all conditional probabilities from both large and small models. In the first step, we compute the transport plan with $|S_z| = K = 6$ and the sequential selection algorithm will select '*be*', which appeared thrice in our samples. We then compute the transport plan with $|S_z| = 3$ and the sequential selection algorithm will select '*liked*'. We then compute the transport plan with $|S_z| = 2$ and the sequential selection algorithm will select '*by*'. Finally, we compute the transport plan with $|S_z| = 1$ and the sequential selection algorithm will not select any of the drafts.

Table 1: Average latency with parallelization along the time axis and batch axis. We report average latency with standard deviation over 1,000 runs using a $97M$ transformer relative to length = 4 and batch = 1 on GPU.

| Relative latency | batch = 1 | batch = 2 | batch = 4 | batch = 8 |
|---|---|---|---|---|
| length = 4 | $1.00 \pm 0.16$ | $1.01 \pm 0.15$ | $1.06 \pm 0.10$ | $1.10 \pm 0.16$ |
| length = 8 | $1.01 \pm 0.18$ | $1.09 \pm 0.25$ | $1.10 \pm 0.09$ | $1.42 \pm 0.4$ |

Table 2: Experimental results on the LM1B dataset. All results are over 1000 test prompts averaged over three different random seeds.

| Algorithm | $K$ | $L$ | Number of decoded tokens per serial call |
|---|---|---|---|
| Baseline | - | - | 1.0 |
| Speculative | 1 | 4 | 2.2 |
| SpecTr | 2 | 4 | 2.4 |
| SpecTr | 4 | 4 | 2.7 |
| SpecTr | 8 | 4 | **3.0** |
| Speculative | 1 | 8 | 2.3 |
| SpecTr | 2 | 8 | 2.6 |
| SpecTr | 4 | 8 | 3.0 |
| SpecTr | 8 | 8 | **3.3** |

phase. Due to the property of the transport plan, we know that $Y$ is always a valid sample from $\mathcal{M}_b(\cdot \mid x^t)$. The overall performance of the algorithm is stated in Theorem 3. We needed to take care in the statement and the proof to deal with the fact that the length $\tau$ of the output sequence $Y^\tau$ is itself a random variable. We defer the proof to the appendix due to limited space.

**Theorem 3.** *Assume all drafts in set $S$ are generated from the small model with input $x^t$, or more precisely, $\forall z \in S$,*

$$\forall i \in [1, L], \qquad z(i) \sim \mathcal{M}_b(\cdot \mid x^t, z^{i-1}). \tag{8}$$

*Let $Y^\tau$ be the output of Algorithm 3 where $\tau$ is the length of the output, and $Z^{\tau+1:L} = (Z(\tau+1), \ldots, Z(L)) \sim \mathcal{M}_b(\underbrace{\cdot, \cdot, \ldots}_{(L-\tau)\,dots} \mid x^t, Y^\tau)$, then it satisfies that $(Y^\tau, Z^{\tau+1:L}) \sim_{\mathrm{prob}} \mathcal{M}_b(\underbrace{\cdot, \cdot, \ldots}_{L\,dots} \mid x^t)$. More precisely, For any length-L sequence $o^L = (o(1), \ldots, o(L)) \in \Omega^L$, we have*

$$\Pr\left((Y^\tau, Z^{\tau+1:L}) = o^L\right) = \Pi_{i=1}^L \mathcal{M}_b(o(i) \mid x^t, o^{i-1}).$$

# 8 Experiments

We evaluate the performance of our algorithm and compare it to speculative decoding by following a recipe provided in [15]. We train decoder-only transformer models on the one-billion language benchmark (LM1B) [3] based on the example provided in the FLAX library [12]. For the draft model, we use a $6M$ parameter transformer model, and for the large model we use a 97M parameter transformer model.

We first provide a verification of the computational model introduced in Section 1 by reporting the latencies of using the large model to compute the probabilistic distributions with parallelization over time and batch axes. As shown in Table 1, the latency stays roughly constant in these setting.

The results of different decoding algorithms are shown in Table 2. The baseline method decodes one token from the large model per serial call, and speculative decoding improves this to $\approx 2.3$. The proposed method SpecTr improves upon speculative decoding and increases the number of decoded tokens per serial call as we increase the number of drafts $K$. We further note that for both Speculative decoding and SpecTr, the number of decoded tokens increases as we increase the block length from 4 to 8. We also note that based on our current implementation, generating the drafts using the small models adds about 10%-15% latency under settings in Table 2. Due to space constraints, we provide additional experiments and details in Appendix F.

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

 # A  Speculative decoding

---

**Algorithm 4** Speculative Sampling SPECSAMPLE.

---

**Input:** Input sequence $x^t$. Access to a small model $\mathcal{M}_s$ and a large model $\mathcal{M}_b$, block length $L$, end
of sequence symbol *eos*.
1: Autoregressively sample $\mathcal{M}_s$ with context $x^t$ to get $L-1$ subsequent samples denoted by
$\tilde{x}_{t+1}, \ldots, \tilde{x}_{t+L-1}$.
2: Let $\tilde{x}_i = x_i$ for $i \leq n$.
3: In parallel compute $p_i = \mathcal{M}_s(\cdot|\tilde{x}^{t+i-1})$ and $q_i = \mathcal{M}_b(\cdot|\tilde{x}^{t+i-1})$ for $1 \leq i \leq L$.
4: **for** $i = 1, \ldots, L-1$ **do**
5:     Compute $Y_i, accept = Algorithm\ 1(p_i, q_i, \tilde{x}_{t+i})$
6:     $x_{t+i} = Y_i$.
7:     **if** $x_{t+i} = eos$ **then**
8:         **Return** $x^{t+i}$
9:     **end if**
10:     **if** $accept = \texttt{True}$ **then**
11:         **Continue**.
12:     **else**
13:         **Return** SPECSAMPLE$(x^{t+i}, \mathcal{M}_s, \mathcal{M}_b, L)$.
14:     **end if**
15: **end for**
16: Draw $x_{t+L}$ from $q_L$.
17: **Return** SPECSAMPLE$(x^{t+L}, \mathcal{M}_s, \mathcal{M}_b, L)$.

---

 # B  Missing proofs in Section 5

 ## B.1  Proof of Lemma 1

 We first prove *monotonicity*. By definition,

$$\alpha_k(p, q) = 1 - \min_{\pi \in \Pi(p^{\otimes k}, q)} \Pr_{X^k, Y \sim \pi}\left(Y \notin S(X^k)\right)$$

$$= \max_{\pi \in \Pi(p^{\otimes k}, q)} \Pr_{X^k, Y \sim \pi}\left(Y \in S(X^k)\right)$$

 Moreover, for any $\pi \in \Pi(p^{\otimes k}, q)$, we can construct $\pi' \in \Pi(p^{\otimes k+1}, q)$ by setting

$$\forall x^{k+1} \in \Omega^{k+1}, y \in \Omega, \pi'(x^{k+1}, y) = \pi(x^k, x(k+1), y)p(x(k+1)),$$

 *i.e.,* adding and independent sample from $p$ to $X^k$.

 Hence we have

$$\alpha_k(p, q) = \max_{\pi \in \Pi(p^{\otimes k}, q)} \Pr_{X^k, Y \sim \pi}\left(Y \in S(X^k)\right)$$

$$= \max_{\pi \in \Pi(p^{\otimes k}, q)} \Pr_{X^{k+1}, Y \sim \pi'}\left(Y \in S(X^k)\right)$$

$$\leq \max_{\pi \in \Pi(p^{\otimes k}, q)} \Pr_{X^{k+1}, Y \sim \pi'}\left(Y \in S(X^{k+1})\right)$$

$$\leq \max_{\pi' \in \Pi(p^{\otimes k+1}, q)} \Pr_{X^{k+1}, Y \sim \pi'}\left(Y \in S(X^{k+1})\right)$$

$$= \alpha_{k+1}(p, q).$$

 Next we prove *consistency*. We start with the case when $\forall x \in \Omega, q(x)/p(x) < \infty$. To prove this, we
 will show that Algorithm 2 with $\gamma_{\max} = \max_{x \in \Omega} q(x)/p(x)$ statisifies

$$\lim_{k \to \infty} \alpha(\pi^{\text{K-SEQ}}_{\gamma_{\max}}) = 1.$$

368    Notice that by Lemma 3 and Theorem 2, $\pi_{\gamma_{\max}}^{\text{K-SEQ}}$ is a valid coupling, and

$$\alpha(\pi_{\gamma_{\max}}^{\text{K-SEQ}}) = 1 - (1 - \beta_{p,q}(\gamma_{\max}))^k,$$

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

**Lemma 3.** *Let*
$$f(\gamma) = 1 - (1 - \beta_{p,q}(\gamma))^k - \gamma\beta_{p,q}(\gamma).$$
*Then we have Let $\gamma^*$ be the solution to Eq. (6). Then when $d_{\mathrm{TV}}(p, q) \in (0, 1)$,*

- *$f(\gamma)$ is monotone in $\gamma$ in $[1, \infty)$;*

- *$\gamma^* \in [1, \min\{k, \max_x \frac{q(x)}{p(x)}\}]$.*

*Proof.* It would enough to prove the followings: (1) $f(\gamma)$ is monotone in $\gamma$ in $[1, \infty)$; (2) $f(1) \geq 0$; (3) $f(k) \leq 0$; (4) $f\left(\max_x \frac{q(x)}{p(x)}\right) \leq 0$.

To see (1), since $\beta_{p,q}(\gamma)$ is decreasing in $\gamma$, so is $1 - (1 - \beta_{p,q}(\gamma))^k$. Moreover, $\gamma\beta_{p,q}(\gamma) = \sum_x \min\{\gamma p(x), q(x)\}$, which is non-decreasing in $\gamma$. Hence we have $1 - (1 - \beta_{p,q}(\gamma))^k - \gamma\beta_{p,q}(\gamma)$ is decreasing.

To see (2), note that when $\gamma = 1$, $\beta_{p,q}(\gamma) = 1 - d_{\mathrm{TV}}(p, q)$. Hence we have

$$1 - (1 - \beta_{p,q}(1))^k = 1 - d_{\mathrm{TV}}(p, q)^k \geq 1 - d_{\mathrm{TV}}(p, q).$$

When $\gamma = k$, (3) holds since for $x \in [0, 1]$, we have $1 - (1 - x)^k \leq kx$. Moreover, when $\gamma = \max_x \frac{q(x)}{p(x)} > 1$, we have $\beta_{p,q}(\gamma) = 1/\gamma$ and (4) holds since

$$1 - (1 - \beta_{p,q}(\gamma))^k = 1 - (1 - 1/\gamma)^k < 1 = \gamma \cdot 1/\gamma.$$

$\square$

Next we prove Theorem 2, we will break the proof into four parts: (1) computation efficiency; (2) $\pi_\gamma^{\text{K-SEQ}}$ is a valid transport plan; (3) acceptance probability; (4) optimality guarantee of $\pi_{\gamma^*}^{\text{K-SEQ}}$.

**Computation efficiency.** Note that the lemma immediately implies that $\gamma^*$ can be computed up to arbitrary accuracy $\delta$ in time $|\Omega| \log(k/\delta)$ using binary search over $[1, k]$.

**Valid transport plan.** We next prove that $\pi_\gamma^{\text{K-SEQ}}$ is a valid transport plan when $\gamma \geq \gamma^*$. By Lemma 3, when $\gamma \geq \gamma^*$, we have $1 - (1 - \beta_{p,q}(\gamma))^k \geq \gamma \beta_{p,q}(\gamma)$. Recall that $p_{\text{acc}} = 1 - (1 - \beta_{p,q}(\gamma))^k$, and

$$\forall x \in \Omega, p^{\text{res}}(x) = \frac{q(x) - \min\left\{p(x), \frac{q(x)}{\gamma}\right\} \frac{p_{\text{acc}}}{\beta_{p,q}(\gamma)}}{1 - p_{\text{acc}}}.$$

$\forall x \in \Omega$, we have

$$\min\left\{p(x), \frac{q(x)}{\gamma}\right\} \frac{p_{\text{acc}}}{\beta_{p,q}(\gamma)} \leq \frac{1 - (1 - \beta_{p,q}(\gamma))^k}{\gamma \beta_{p,q}(\gamma)} q(x) \leq q(x),$$

this implies $p^{\text{res}}(x) \geq 0$ for all $x \in \Omega$. Moreover,

$$\sum_{x \in \Omega} p^{\text{res}}(x) = \sum_{x \in \Omega} \frac{q(x) - \min\left\{p(x), \frac{q(x)}{\gamma}\right\} \frac{p_{\text{acc}}}{\beta_{p,q}(\gamma)}}{1 - p_{\text{acc}}} = 1.$$

Hence $p^{\text{res}}$ is a valid distribution. It remains to show that the marginal of $Y$ is $q$. We first compute the probability of the output $Y = x$. Note that probability that $Y = X_1$ is

$$p(X_1) \min\left(1, \frac{q(X_1)}{\gamma p(X_1)}\right) = \min\left(p(X_1), \frac{q(X_1)}{\gamma}\right).$$

Hence

$$\Pr(Y = X_1 = x) = \min\left(p(x), \frac{q(x)}{\gamma}\right).$$

Therefore,

$$\Pr(Y = X_1) = \sum_x \min\left(p(x), \frac{q(x)}{\gamma}\right) = \beta(\gamma).$$

Similarly, probability that

$$\Pr(Y = X_2 = x) = \Pr(Y \neq X_1) \Pr(Y = X_2 | Y \neq X_1) = (1 - \beta_{p,q}(\gamma)) \min\left(p(x), \frac{q(x)}{\gamma}\right).$$

Hence,

$$\begin{aligned}
&\Pr\left(Y = x, \text{one of } X^k \text{ is accepted}\right) \\
&= \sum_{i=0}^{k-1} \Pr\left(X_1, \ldots, X_i \text{ are rejected}, X_{i+1} \text{ is accepted, and } X_{i+1} = x\right) \\
&= \sum_{i=0}^{k-1} (1 - \beta_{p,q}(\gamma))^i \cdot p(x) \cdot \min\left\{1, \frac{q(x)}{p(x)\gamma}\right\} \\
&= \min\left\{p(x), \frac{q(x)}{\gamma}\right\} \cdot \sum_{i=0}^{k-1} (1 - \beta_{p,q}(\gamma))^i \\
&= \min\left\{p(x), \frac{q(x)}{\gamma}\right\} \frac{1 - (1 - \beta_{p,q}(\gamma))^k}{\beta_{p,q}(\gamma)}
\end{aligned}$$

Summing over all symbols $x$ yields

$$\Pr(\text{one of } X^k \text{ is accepted}) = \sum_x \min\left\{p(x), \frac{q(x)}{\gamma}\right\} \frac{1 - (1 - \beta_{p,q}(\gamma))^k}{\beta_{p,q}(\gamma)} = 1 - (1 - \beta_{p,q}(\gamma))^k.$$

Hence we have

$$\Pr\left(Y = x\right) = \Pr\left(Y = x, \text{one of } X^k \text{ is accepted}\right) + (1 - \text{one of } X^k \text{ is accepted})p^{\text{res}}(x)$$

$$= \min\left\{p(x), \frac{q(x)}{\gamma}\right\} \frac{1 - (1 - \beta_{p,q}(\gamma))^k}{\beta_{p,q}(\gamma)}$$

$$+ \left(1 - (1 - \beta_{p,q}(\gamma))^k\right) \frac{q(x) - \min\left\{p(x), \frac{q(x)}{\gamma}\right\} \frac{1 - (1 - \beta_{p,q}(\gamma))^k}{\beta_{p,q}(\gamma)}}{1 - (1 - \beta_{p,q}(\gamma))^k}$$

$$= q(x).$$

**Acceptance probability.** The acceptance probability holds since

$$\alpha(\pi_\gamma^{\text{K-SEQ}}) \geq \Pr(\text{one of } X^k \text{ is accepted}) = 1 - (1 - \beta_{p,q}(\gamma))^k.$$

**Optimality guarantee of $\pi_{\gamma^*}^{\text{K-SEQ}}$.** It can be seen that $\beta(\gamma)$ is decreasing in $\gamma$, and so is $1 - (1 - \beta_{p,q}(\gamma))^k$. Hence we have

$$\alpha(\pi_{\gamma^*}^{\text{K-SEQ}}) \geq 1 - (1 - \beta_{p,q}(\gamma^*))^k \geq 1 - (1 - \beta_{p,q}(k))^k = c_k(p,q) \cdot \min\{kp(x), q(x)\},$$

where

$$c_k(p,q) = \frac{1 - (1 - \beta_{p,q}(k))^k}{k\beta_{p,q}(k)} \in [1 - (1 - 1/k)^k, 1).$$

The inclusion holds since $f(x) = \frac{1 - (1-x)^k}{kx}$ in monotonically decreasing when $x \in (0, 1/k]$ and $f(1/k) = 1 - (1 - 1/k)^k$, $\lim_{x \to 0^+} f(x) = 1$.

Moreover, $\forall x \geq 0, kx \geq 1 - (1 - x)^k$. Hence we have

$$\alpha(\pi_{\gamma^*}^{\text{K-SEQ}}) \geq \left(1 - (1 - 1/k)^k\right) \cdot \min\{kp(x), q(x)\}$$

$$\geq \left(1 - (1 - 1/k)^k\right) \min\{1 - (1 - p(x))^k, q(x)\}$$

$$\geq \left(1 - (1 - 1/k)^k\right) \alpha_k(p,q),$$

where the last inequality is due to the upper bound in Theorem 1 with $\Omega_0 = \Omega$.

## C.2  Proof of Theorem 3

We prove the theorem via induction. When $L = 1$, $\Pr\left(\tau = 1\right) = 1$, the theorem follows directly since $f_\pi$ in Algorithm 3 is a valid transport plan. Suppose the theorem holds for $L = \ell \geq 1$, we next prove that it holds for $L = \ell + 1$. Let $\bar{Y}^{\tau'}$ be the output sequence when $L = \ell + 1$ and $\bar{Z}^{\tau'+1:\ell+1}$ be the subsequent samples from $\mathcal{M}_b$. Note that compared to the case when $L = \ell$, extending the block length of the tree by one only changes the probability of $Y^\tau$ when $\tau = L$, *i.e.*, $\forall j < \ell$ and length-$j$ sequence $o^j \in \Omega^j$, we have

$$\Pr\left(Y^j = o^j, \tau = j\right) = \Pr\left(\bar{Y}^j = o^j, \tau' = j\right)$$

For any length-$\ell$ sequence $o^\ell$, let

$$\delta(o^\ell) := \Pr\left(Y^\ell = o^\ell, \tau = \ell\right) - \Pr\left(\bar{Y}^\ell = o^\ell, \tau' = \ell\right).$$

Then by definition, we have

$$\delta(o^\ell) = \sum_{o(\ell+1) \in \Omega} \Pr\left(Y^{\ell+1}\right) = (o^\ell, o(\ell+1)), \tau = \ell + 1)$$

For any length-$(\ell+1)$ sequence $o^{\ell+1} \in \Omega^{\ell+1}$, we have

$$\Pr\left((\bar{Y}^j, \bar{Z}^{\tau'+1:\ell+1}) = o^{\ell+1}\right) \tag{9}$$

$$= \sum_{j=1}^{\ell-1} \Pr\left(\bar{Y}^j = o^j, \tau' = j\right) \mathcal{M}_b(o^{j+1:\ell+1} \mid x^n, o^j)$$

$$+ \Pr\left(\bar{Y}^\ell = o^\ell, \tau' = \ell\right) \mathcal{M}_b(o(\ell+1) \mid x^n, o^\ell) + \Pr\left(\bar{Y}^{\ell+1} = o^{\ell+1}, \tau = \ell+1\right) \tag{10}$$

$$= \sum_{j=1}^{\ell-1} \Pr\left(Y^j = o^j, \tau' = j\right) \mathcal{M}_b(o^{j+1:\ell+1} \mid x^n, o^j)$$

$$+ \left(\Pr\left(Y^\ell = o^\ell, \tau' = \ell\right) - \delta(o^\ell)\right) \mathcal{M}_b(o(\ell+1) \mid x^n, o^\ell) + \Pr\left(\bar{Y}^{\ell+1} = o^{\ell+1}, \tau = \ell+1\right) \tag{11}$$

$$= \mathcal{M}_b(o^{\ell+1} \mid x^n) - \delta(o^\ell) \mathcal{M}_b(o(\ell+1) \mid x^n, o^\ell) + \Pr\left(\bar{Y}^{\ell+1} = o^{\ell+1}, \tau = \ell+1\right). \tag{12}$$

Hence it would enough to show that

$$\delta(o^\ell) \mathcal{M}_b(o(\ell+1) \mid x^n, o^\ell) = \Pr\left(\bar{Y}^{\ell+1} = o^{\ell+1}, \tau = \ell+1\right) \tag{13}$$

Note that the event $\bar{Y}^{\ell+1} = o^{\ell+1}, \tau' = \ell+1$ only happens when $o^\ell$ are all accepted samples from $\mathcal{M}_s$ in the sampling process and when proceeding, the next obtained token is $o(\ell+1)$.

On the other hand, the $\delta(o^\ell)$ is the probability of the event that the sampling process stops at $o^\ell$ when $L = \ell$ and proceeds when $L = \ell+1$, which, by definition of the algorithm, happens if and only if $o^\ell$ are all accepted samples from $\mathcal{M}_s$. Moreover, when proceeding, since $f_\pi$ is a valid transport plan, we have that the next sample is generated from $\mathcal{M}_b(\cdot \mid x^t, o^\ell)$. And hence Eq. (13) holds.

This concludes the proof.

# D  Comparisons between OTM-$k$ and K-SEQ

## D.1  Examples where the approximate algorithm is optimal

In this section, we show that for the example in Figures 1, K-SEQ achieves the optimal acceptance accuracy. In this case, $p = U(d)$ and $q = U(d/r)$. Recall that the optimal acceptance probability is

$$\alpha_k(U(d), U(d/r)) = 1 - (1 - 1/r)^k.$$

For $U(d)$ and $U(d/r)$, we have

$$\beta(\gamma) = \sum_{x \in [d]} \min\{p(x), q(x)/\gamma\} = \frac{1}{\max\{r, \gamma\}}.$$

And hence solving $1 - (1 - \beta(\gamma))^k = \gamma\beta(\gamma)$ gives $\gamma^* = r(1 - (1 - 1/r)^k)$. And be Theorem 2, we have

$$\alpha(\pi_{\gamma^*}^{\text{K-SEQ}}) \geq 1 - (1 - \beta(\gamma^*))^k = 1 - (1 - 1/r)^k.$$

And the equality holds since this an upper bound for any coupling.

## D.2  Gap between OTM-$k$ and K-SEQ

To see how OTM-$k$ and K-SEQ compare in general, we numerically compute the acceptance probability for a pair of compressed conditional distributions. We feed the prompt

*"He said he also has asked prosecutors to"*

to both large and small models used in Section 8 and obtain the conditional distributions $p, q$. To make the computation feasible for OTM-$k$, we take the set of top 10 elements from $p, q$ respectively and set the support $S$ to be the union of the two sets. Then we set $p'$ and $q'$ to be the normalized distribution of $p$ and $q$ over the set $S$.

We then numerically compute the acceptance probability for the optimal transport solution in Section 5 and the approximate solution in Section 6 with different $k$'s. The result in shown in Fig. 5. When $k = 1$, the acceptance probability is equal to $1 - d_{\text{TV}}(p', q')$ for both solutions. The acceptance probability increases for both methods as $k$ increases and there exists a gap between the optimal and approximate solution. We would expect the gap to exist for general conditional distributions from language models. We leave exploring computationally efficient ways to close this gap as an interesting future direction.

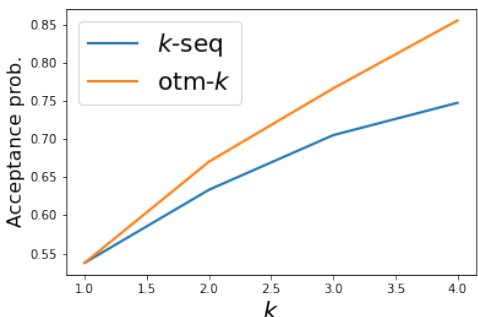

Figure 5: Acceptance probability comparison OTM-$k$ and K-SEQ with compressed conditional distributions.

# E    Construct a candidate set by sampling from a prefix-tree