# OpenReview forum: "SpecTr: Fast Speculative Decoding via Optimal Transport"
_NeurIPS.cc/2023/Conference — NeurIPS 2023 poster_

### Official Review · Reviewer_DdMT · 2023-07-03

**Soundness:** 2 fair
**Presentation:** 2 fair
**Contribution:** 2 fair
**Rating:** 4
**Confidence:** 3

**Summary:**

The paper proposes SpecTr, a new speculative decoding framework for efficient Transformer and LLM decoding.
SpecTr uses a combination of a small, fast model for generating draft samples and a larger, more accurate model for scoring and validating them in parallel.
Compared to the prior works, the paper presents a formulation of the speculative decoding process through the lense of optimal transport theory and demonstrates the potential for further acceleration by generating multiple drafts in parallel along the batch axis.
The authors offer an approximate solution to efficiently solve the optimal transport problem at scale, resulting in significant latency improvements compared to baseline methods and previous speculative decoding approaches.

**Strengths:**

The paper presents theoretical justifications for the methods that it proposes. Furthermore, it is the first attempt to incorporate parallelization along batch into the speculative sampling framework for better efficiency.

**Weaknesses:**

1. While the paper proposes theoretical justifications for the proposed method, it rather lacks proper evaluation. Since accelerating decoding processes is a very practical area, the author should provide a more thorough analysis of the end-to-end latency and text generation performance. In particular:
   * (a) The paper lacks an evaluation of the impact of the proposed method on text generation performance. Tables 2 and 4 in the paper only focus on latency, and there is no comparison in terms of text generation quality (e.g. measured by metrics like BLEU scores, etc.) compared to the baseline or prior speculative methods. Latency numbers without these performance metrics make it difficult to assess the effectiveness of the proposed method.
   * (b) The number of decoded tokens per serial call (in Table 2) is not directly indicative of the actual latency and does not necessarily represent the actual speedup. That is, it serves as a proxy measurement that may or may not yield the same degree of latency improvement. This is because there are factors such as framework overheads [1], hardware utilization, and others that can potentially impact the latency. For instance, it is difficult to state if processing 3 tokens (as in Table 2) in a single call is truly 3 times more latency efficient than processing them sequentially in 3 separate calls. Therefore, the claimed 3X speedup, and the additional 1.36X improvement over speculative decoding mentioned in the abstract, can be misleading.
   * (c) It is unclear whether the proposed method introduces any additional run-time overhead compared to the prior speculative sampling method. If there is indeed extra overhead involved, it could potentially reduce the gap between the two methods, making the improvements presented in Table 2 less significant (thereby making the 1.36X improvement over the prior method misleading as well).
   * (d) The latency overhead of running smaller models concurrently remains uncertain, since Table 2 only presents the efficiency of running the large model. For instance, a longer sequence length (L) can decrease runtime costs for the large model by increasing the number of tokens decoded per call as stated in the table. However, at the same time, it also increases the small model's cost as more tokens are generated but rejected/discarded. The paper lacks discussions around this point. While the authors presented the latency overhead of 10-15%, it is not clear under which particular setting it was measured and how the hyperparameters (K and L) affects this value.
   * (e) Taken together, the authors should provide the end-to-end inference latency and the text-generation quality/performance measure to make the evaluation convincing.

2. Including a concluding paragraph in the paper would enhance its professionalism. The authors should at least wrap up the paper with a one-paragraph summary.

[1] The Framework Tax: Disparities Between Inference Efficiency in Research and Deployment, https://arxiv.org/pdf/2302.06117.pdf

**Questions:**

N/A

**Limitations:**

Please see the weakness section

---

> ### Author Rebuttal · Authors · 2023-08-09
>
> Thanks for agreeing on the novelty of the algorithm and the theoretical justification. Below we address the concerns on the evaluation of our method.
>
> #### ***[Quality of the final outputs.] (response to 1(a))***
> (Identical to the same comment in the global response, provided here for completeness.)  We would like to clarify that one of the biggest advantages of our proposed acceleration methods is that there is **provably** no drop in performance because our algorithm guarantees that the final outcome is a statistical draw from the large model. This holds for both the optimal solution from exactly solving the linear program, and the approximate solution $k$-seq. The “approximation” part in $k$*-seq* comes from the non-optimal acceptance probability, which leads to less decoded tokens per serial call and a hit in latency compared to the optimal solution.
>
> More precisely, SpecTr guarantees that the final output sequence follows the exact same distribution as the output sequence from the large model as long as the token-level algorithm is a valid coupling between $p^{\otimes k}$ and $q$ where $p,q$ are the conditional distributions on the next token from the small and large model respectively. Hence all metrics such as BLEU, avg. sentence likelihood, will be *neutral*. The formal statement is stated in Theorem 3 of the submission. The guarantee is the same as what is claimed in previous speculative decoding methods.
>
> We will add more discussions in the revised version and improve the theorem statement to make this fact clearer.
>
> #### ***[End-to-end latency improvement.] (response to 1(b-e))***
> (Mostly the same with comment in the global response, provided here for completeness.) We agree that all factors mentioned by the reviewer would affect the effectiveness of the proposed approach in practical systems, and it is important to implement the method and report end-to-end latency comparisons including the delays caused by system overheads. To further demonstrate the effectiveness of our proposed approach, we conduct experiments on the state-of-the-art PALM-2 models [1] with PALM-2-Gecko and PALM-2-Bison (where Bison is a larger model) as the small model and large model, respectively. We report end-to-end (wall clock) latency comparisons between regular decoding, speculative decoding, and SpecTr. This includes the time to (parallelly) draw drafts from the small model, the time to verify the drafts with the large model, latency introduced by running the sequential rejection algorithm, and other system overheads. See the attached PDF file in the global response for detailed numbers. While we do see a smaller wall clock speed-up compared to the number of decoded tokens per serial call, our proposed method still achieves significant improvement in wall clock latency with respect to baseline decoding and speculative decoding. When $K =8$ and $L=8$, our relative wall clock speed-up over baseline is 2.13x (in contrast to 1.56x for speculative decoding over baseline), a further 1.37x improvement over speculative decoding.
>
> We will add additional experimental results in the final version.
>
> [1] Google   AI. Introducing   PaLM   2, 2023. https://blog.google/technology/ai/google-palm-2-ai-large-language-model/.

---

### Official Review · Reviewer_jweu · 2023-07-07

**Soundness:** 3 good
**Presentation:** 3 good
**Contribution:** 3 good
**Rating:** 6
**Confidence:** 3

**Summary:**

The paper studies speculative decoding, a technique to increase inference efficiency in a large autoregressive model by sampling a set of candidate tokens (a *draft*) from a smaller (thus, faster) model, which is then scored according to the conditional distribution of the original larger model. The scores of the larger model are obtained in parallel, thus potentially resulting in a significant speedup depending on how well the smaller model's conditional distribution is close to the larger model's. The main contributions are twofold. First, the authors cast the speculative decoding problem as a maximal coupling problem in optimal transport. Then, they propose to use multiple drafts (i.e. multiple sets of tokens completing the context), which is solvable in exponential time in the number of drafts. Thus, a novel algorithm is proposed that runs linearly in the number of drafts and provably solves the problem up to a $(1-1/e)$ factor of the optimal acceptance probability.

**Strengths:**


1. To my knowledge, translating the speculative decoding problem in the context of optimal transport is not very surprising but novel. Furthermore, by connecting the problem to the field of optimal transport, the paper opens new possibilities for research, due to the maturity of the theory of optimal transport, which the authors actually use to derive an extension of the original speculative decoding algorithm to allow for multiple drafts and discuss its feasibility in terms of algorithmic complexity.

2. The proposed algorithm to approximately compute the transport plan (Algorithm 2) nicely follows the problem casting performed earlier and is backed by a sound theory (Theorem 2) describing how the solution (in terms of acceptance probability) compares to the intractable case of exact solution in exponential time. I have quickly skimmed through the appendix and although not an expert in OT, the theory seems correct.

**Weaknesses:**


1. The idea of extending the speculative decoding algorithm in [15]  to multiple drafts is simple and elegant, but sometimes the authors add some unnecessary details. For instance, it seems trivial and very intuitive that by having multiple drafts, the acceptance probability increases with the number of drafts (hence Lemma 1 seems unnecessary).

2. The authors test their proposed method solely on LM1B. I would have liked to see a larger set of experiments, covering at least a more significant subset of those performed in [1].

3. Algorithm 1 is wrong, as it always returns at line 7. I guess the authors want it to return at line 5 as well. Also, some quantities are not defined yet when Alg.1 is introduced (such as $\mathcal{X}$), which complicates a bit the readability.

Minor:

3. In Figures 1,2,3 the size of ticks, labels, and lines should be increased.

4. Typos: line 251: "Gvien". Line 217-218 the sentence "To control the probability of accepting an $x \in \Omega$ with probability larger than $q(x)$." does not seem to be grammatically correct. Line 44 "a several contexts".  Line 120: "an discrete"

[1] Yaniv Leviathan, Matan Kalman, and Yossi Matias. Fast inference from transformers via speculative decoding. arXiv preprint arXiv:2211.17192, 2022.

**Questions:**

None.

**Limitations:**

The authors address the limitations. I do not see any potential negative societal impact.

---

> ### Author Rebuttal · Authors · 2023-08-09
>
> We thank the reviewer for the positive comments, pointing out the typos, and suggesting ways to improve the paper. We will incorporate them in the final version. Please see inline replies below.
>
> #### ***[Lemma 1 seems unnecessary]***
> We agree that the monotonicity part of  Lemma 1 follows from definitions, but the consistency part needs proof. We will clarify in the subsequent version.
>
> #### ***[Larger set of experiments]***
> For the rebuttal, we have evaluated our method on state-of-the-art PALM-2 models and report end-to-end latency improvements. Please see details in the global response. We will add additional experimental results in the final version.
>
> #### ***[Algorithm 1 always returns at Line 7]***
> Thank you for pointing out the typo. As you mention, Line 5 of the algorithm should say Return Y = X and accept = True. Will add other definitions in the Algorithm.
>
> #### ***[Typos & suggestions on font size]***
> Thanks for these suggestions. We will incorporate them.

---

### Official Review · Reviewer_hEU2 · 2023-07-14

**Soundness:** 3 good
**Presentation:** 3 good
**Contribution:** 3 good
**Rating:** 7
**Confidence:** 3

**Summary:**

This paper presents speculative decoding, wherein a smaller language model is used to approximately sample from a large one, akin to
a type of accept/reject MCMC sampler.

The idea is that it is slow to sequentially sample from a large model of interest, but joint probabilities can be computed in parallel across the time dimension. At the same time, we could have access to many parallel copies of a smaller not-as-good model, and it's quicker to generate e.g. 100 tokens from the small model than to sample from the large one. We generate e.g. 100 tokens for K such smaller models, and accept one of the K sequences with some probability.

The paper is mostly a theory paper that gives bounds on the transport cost, between the large model distribution and the sampling distribution resulting from the accept/reject sampling composed with proposing from the smaller models.

On the plus side, the paper is a theory paper that I think has some nice implications for practitioners. I would love to see some more empirical work exploring this. The quality of the math / theory / bounds is high and the authors did a nice job breaking down the OT parts for the typical ML language modeling reader.

On the minus side, while a light set of experiments is totally fine for a theory paper, I believe most readers of the paper would benefit more if there were also some initial results of the actual sample quality from the presented algorithm. There seem to be results on latency and accept/reject probability (and there's some theory connecting accept/reject probability to closeness of sampled distribution to desired large model distribution), but there seem to be no direct results evaluating the samples resulting from the new algorithm.

In short, I tentatively accept but think the minus needs to be addressed and will change the score if there is no discussion or clarification on this. Note that light evaluation is totally fine since the theory is good. And it is even okay if the quality of the samples according to any of the metrics is possibly below competitive, since there is always room to improve these algorithms once they are established.


**Strengths:**

- novel algorithm for quicker sampling making use of smaller models and MCMC-like concepts to sample from a larger model
- nice framing in terms of optimal transport that I think the typical ML + LM reader could take something positive away from
- rigorous characterization theoretically of the algorithm in terms of transport cost bounds, accept/reject rates, etc.

**Weaknesses:**

- Table mismatch: e.g. referenced table 8 in main text but no table 8 in main text or appendix
- Most noticeably, there seem to be no direct reports of quality of samples from the resulting approximate algorithm. There are tables for latency and for accept/reject rates (the latter is in appendix but could probably go in main text). And there is theory characterizing that the transport cost will not be too bad, etc. But, there are no direct metrics quantifying the original large model samples vs the proposed algorithm's samples (those produced by using the smaller models' sampling and larger models' scoring). Seems like any of the basic LM metrics would be warranted here

Glad to be corrected if I have misunderstood.

Minor:
- Small type "Gvien" in source line 251 of the PDF.


**Questions:**

See Strengths/Weaknesses

**Limitations:**

Yes

---

> ### Author Rebuttal · Authors · 2023-08-09
>
> We thank the reviewer for the positive comments on the novelty of the algorithm and soundness of the theory. Please see replies to specific questions below.
>
> #### ***[Quality of the samples]***
> (Identical to the same comment in the global response, provided here for completeness.) We would like to clarify that one of the biggest advantages of our proposed acceleration methods is that there is **provably** no drop in performance because our algorithm guarantees that the final outcome is a statistical draw from the large model. This holds for both the optimal solution from exactly solving the linear program, and the approximate solution $k$-seq. The “approximation” part in $k$*-seq* comes from the non-optimal acceptance probability, which leads to less decoded tokens per serial call and a hit in latency compared to the optimal solution.
>
> More precisely, SpecTr guarantees that the final output sequence follows the exact same distribution as the output sequence from the large model as long as the token-level algorithm is a valid coupling between $p^{\otimes k}$ and $q$ where $p,q$ are the conditional distributions on the next token from the small and large model respectively. Hence all metrics such as BLEU, avg. sentence likelihood, will be *neutral*. The formal statement is stated in Theorem 3 of the submission. The guarantee is the same as what is claimed in previous speculative decoding methods.
>
> We will add more discussions in the revised version and improve the theorem statement to make this fact clearer.
>
> #### ***[Scale of the empirical evaluation]***
> We agree that performing a more complete set of experimental evaluations would provide more evidence on the effectiveness of the method. For the rebuttal, we also evaluate our proposed method on the state-of-the-art PALM-2 models and report end-to-end latency improvements with respect to baseline autoregressive decoding and speculative decoding. Please see details in the global response and Table 1 in the attached PDF. We will add additional experimental results in the final version.
>
> #### ***[Table 8]***
> Thanks for catching this! Table 8 was a latex error on our part. The experimental section refers to Tables 1, 3 (wall clock for large and small models) and Table 2 (Results on LM1B dataset).

---

### Official Review · Reviewer_TBLD · 2023-07-26

**Soundness:** 3 good
**Presentation:** 3 good
**Contribution:** 3 good
**Rating:** 6
**Confidence:** 2

**Summary:**

This paper proposes a novel and efficient decoding algorithm for autoregressive large language model, SpecTr, which is an extension of speculative decoding. Given a large model M_b, it can only output one word at a time when it decodes, however, the cost of a serial call is quite expensive. The previous method speculative decoding can alleviate the slow decoding problem, which uses a much smaller model M_s to decode a single segment of length L and allows M_b to calculate its likelihood in parallel. The optimal transport plan between the distributions of the two models is used to determine whether to accept a certain part of the segment, which makes sure the samples has the same distribution as M_b. This approach has the advantage of reducing the number of serial calls of M_b, improving decoding efficiency. However, if the acceptance probability is too low, the reduction may not be significant.

The SpecTr algorithm allows to utilize K i.i.d. segments sampled by M_s to enhance acceptance probability. Concretely, the previous optimal transport problem is modified to transport the K-product distribution of M_s to the distribution of M_b. Since solving the new problem has an exponential cost with respect to K, the paper introduces the k-sequential selection algorithm with a 1-1/e approximation ratio with an appropriate cost.

In numerical experiments, they use an M_b model with 97M parameters and an M_s model with 6M parameters tested on LM1B, speculative decoding achieves an average of decoding 2.3 tokens per serial run of M_b, while the proposed method in this paper can decode 3 tokens, showing a significant improvement.


**Strengths:**

The paper is well-written and self-contained, making it accessible even to someone like me who is not familiar with the related work on language model decoding. The proposed algorithm, SpecTr, combines the practical need for parallelization of large models in real-world scenarios. Moreover, it is mathematically concise, intuitive, and insightful. The paper also provides a basic theoretical analysis of the properties of the algorithm. The numerical experiments show a significance improving of their methods.
I think this is a good submission.


**Weaknesses:**

The numerical experiments on real datasets seem insufficient, as they only compare with speculative decoding when K=1, which may not be fair: using larger K would require more computational resources. It would be worth considering if there is a more equitable way to compare the methods, e.g. extend speculative decoding to use more resources in some crude way.

**Questions:**

The first line of Algorithm 1:
Input: ... "X~iid p" is confusing, since you have only one sample here, "iid" is unnecessary.

---

> ### Author Rebuttal · Authors · 2023-08-09
>
> We thank the reviewer for the positive comments on the novelty of the algorithm and theory, and acknowledging that SpectTr is mathematically concise, intuitive, and insightful.
>
> #### ***[Fair comparison with speculative decoding]***
> We would like to point out that one of the main contributions of our work is to relate speculative decoding to the theory of optimal transport, which allows for the use of $k$ draws from the small model. We are unaware of other intuitive baselines on how to use more resources to speed-up speculative decoding due to the subtly involved in ensuring that the final sequence is still a *valid statistical draw from the large model*, which is our main contribution in this paper. We would be happy to conduct comparisons if we are missing some baseline extensions that you may have in mind.
>
> #### ***[Typo]***
> Thanks for catching; we will fix it.

---

### Author Rebuttal · Authors · 2023-08-09

We thank all reviewers for their detailed reading and encouraging comments about our submission. We are delighted to read the reviewers’ acknowledgement of the novelty of the algorithm (reviewer TBLD, hEU2, jweu, DdMT) and the soundness of the theory (reviewer hEU2, jweu). We will incorporate their suggestions to improve the presentation in future revisions of the paper.  Below we address a few common concerns raised by the reviewers. Each reviewer's individual questions will be answered in separate responses.

#### ***[End-to-end latency improvement.]***
We agree that it is important to implement the method and report end-to-end latency comparisons including the delays caused by system overheads. To further demonstrate the effectiveness of our proposed approach, we conduct experiments on the state-of-the-art PALM-2 models [1] with PALM-2-Gecko and PALM-2-Bison (where Bison is a larger model) as the small model and large model, respectively. We report end-to-end (wall clock) latency comparisons between regular decoding, speculative decoding, and SpecTr. This includes the time to (parallelly) draw drafts from the small model, the time to verify the drafts with the large model, latency introduced by running the sequential rejection algorithm, and other system overheads as suggested by reviewer DdMT. See the attached PDF file for detailed numbers. While we do see a smaller wall clock speed-up compared to the number of decoded tokens per serial call, our proposed method still achieves significant improvement in wall clock latency with respect to baseline decoding and speculative decoding. When $K =8$ and $L=8$, our relative wall clock speed-up over baseline is 2.13x (in contrast to 1.56x for speculative decoding over baseline), a further 1.37x improvement over speculative decoding.

We will add additional experimental results in the final version.

#### ***[Quality of the final outputs.]***
We would like to clarify that one of the biggest advantages of our proposed acceleration methods is that there is **provably** no drop in performance because our algorithm guarantees that the final outcome is a statistical draw from the large model. This holds for both the optimal solution from exactly solving the linear program, and the approximate solution $k$-seq. The “approximation” part in $k$-seq comes from the non-optimal acceptance probability, which leads to less decoded tokens per serial call and a hit in latency compared to the optimal solution.

More precisely, SpecTr guarantees that the final output sequence follows the exact same distribution as the output sequence from the large model as long as the token-level algorithm is a valid coupling between $p^{\otimes k}$ and $q$ where $p,q$ are the conditional distributions on the next token from the small and large model respectively. Hence all metrics such as BLEU, avg. sentence likelihood, will be *neutral*. The formal statement is stated in Theorem 3 of the submission. The guarantee is the same as what is claimed in previous speculative decoding methods.

We will add more discussions in the revised version and improve the theorem statement to make this fact clearer.

[1] Google   AI. Introducing   PaLM   2, 2023. https://blog.google/technology/ai/google-palm-2-ai-large-language-model/.

---

### Author Response · Authors · 2023-08-21

Dear Reviewers and ACs,

Hope the rebuttal has addressed your questions in the review, including questions about **wall clock numbers** (see pdf) and the **quality of the samples** (which is *provably* statistically indifferent from the base model). As the end of discussion period approaches, we would like to take the last chance to resolve your further concerns if any.

Thanks!\
Authors

---

### Decision · Program_Chairs · 2023-09-21

**Decision:**

Accept (poster)

**Comment:**

This paper presents speculative decoding, wherein a smaller language model is used to approximately sample from a large one, akin to a type of accept/reject MCMC sampler. This is a borderline paper. The AC had a thorough discussion with the SAC and the SAC agreed on the decision.